# *Parabacteroides distasonis* ameliorates insulin resistance via activation of intestinal GPR109a

Yonggan Sun[1,2,3,5], Qixing Nie[1,2,3,5], Shanshan Zhang[1,2,3], Huijun He[1,2,3], Sheng Zuo[1,2,3], Chunhua Chen[1,2,3], Jingrui Yang[1,2,3], Haihong Chen[1,2,3], Jielun Hu[1,2,3], Song Li[1,2,3], Jiaobo Cheng[1], Baojie Zhang[1], Zhitian Zheng[1], Shijie Pan[1], Ping Huang[4], Lu Lian[4] & Shaoping Nie[1,2,3] ✉

Gut microbiota plays a key role in insulin resistance (IR). Here we perform a case-control study of Chinese adults (ChiCTR2200065715) and identify that *Parabacteroides distasonis* is inversely correlated with IR. Treatment with *P. distasonis* improves IR, strengthens intestinal integrity, and reduces systemic inflammation in mice. We further demonstrate that *P. distasonis*-derived nicotinic acid (NA) is a vital bioactive molecule that fortifies intestinal barrier function via activating intestinal G-protein-coupled receptor 109a (GPR109a), leading to ameliorating IR. We also conduct a bioactive dietary fiber screening to induce *P. distasonis* growth. *Dendrobium officinale* polysaccharide (DOP) shows favorable growth-promoting effects on *P. distasonis* and protects against IR in mice simultaneously. Finally, the reduced *P. distasonis* and NA levels were also validated in another human type 2 diabetes mellitus cohort. These findings reveal the unique mechanisms of *P. distasonis* on IR and provide viable strategies for the treatment and prevention of IR by bioactive dietary fiber.

Insulin resistance (IR) is a physiological state in which insulin sensitivity decreases in target organs such as skeletal muscle, liver, and white adipose tissue, reducing glucose uptake and utilization, resulting in increased compensatory insulin secretion[1]. IR is characteristic of the subject with type 2 diabetes mellitus (T2DM), an epidemic projected to rise even further to ~700 million by 2045[2–4]. IR is caused by a combination of genetic and environmental factors, but its pathogenesis remains unclear[1,5]. Currently, it is widely accepted that obesity and chronic inflammation are important contributing factors to the development of IR[6,7]. A long-term high-fat-diet (HFD) induces dysbiosis of gut microbiota and disrupts intestinal barrier function, leading to bacterial endotoxemia leaking into the blood and activate toll-like receptor 4 (TLR-4) in liver or fat to induce tumor necrosis factor-

alpha (TNF-α) production[6,8–10]. TNF-α directly affects insulin signaling transduction, which further impedes glucose uptake by cells and induces IR[11].

Gut bacteria and their metabolites can impact host IR in multiple ways. In recent years, studies have demonstrated that dysbiosis of the gut microbiota is inextricably linked to the development and progression of IR[12]. Individuals with IR possessed increased branched-chain amino acid (BCAA) biosynthesis potential, along with the increased abundance of BCAAs-producing bacterium, among which *Prevotella copri* and *Bacteroides vulgatus* are identified as the main species driving the biosynthesis of BCAAs, leading to IR[13]. An increased level of bacteria-derived imidazole propionate deteriorates IR by activating the mechanistic target of rapamycin complex 1 (mTORC1)[14].

[1]State Key Laboratory of Food Science and Resources, Nanchang University, Nanchang, China. [2]China-Canada Joint Lab of Food Science and Technology, Nanchang University, Nanchang, China. [3]Key Laboratory of Bioactive Polysaccharides of Jiangxi Province, Nanchang University, Nanchang, China. [4]Department of Nutrition, the First Affiliated Hospital of Nanchang University, Nanchang, China. [5]These authors contributed equally: Yonggan Sun, Qixing Nie. ✉e-mail: spnie@ncu.edu.cn

In contrast, supplementation with *Lactobacillus reuteri* increased levels of aryl hydrocarbon receptor (AhR) ligand, leading to improved IR in mice by enhancing intestinal barrier function[15]. Therefore, modulating the gut microbiota and enhancing intestinal barrier function are effective strategies for improving IR. Despite mounting evidence that T2DM is linked to the gut microbiota[16–18], few studies have examined IR and gut microbiota association. Moreover, the exacerbation or improvement of IR is linked to different gut microbiota[19], and further investigation is warranted to comprehend the intricate relationship between them.

Dietary fiber, recognized as the seventh major nutrient in humans, is widely derived from daily foods[20]. Dietary fiber cannot be degraded by host-derived enzymes. However, it can be degraded by the gut microbiota and play an essential role in modulating the gut microbial ecosystem[21,22]. Several studies indicate that dietary fiber can relieve IR by enhancing the growth of specific gut bacteria, resulting in the production of microbiota-derived metabolites[23]. A randomized clinical study revealed that short-chain fatty acids (SCFAs)-producing strains in the gut selectively promoted by additional dietary fiber intake alleviate IR in patients with T2DM[24]. Simultaneously, dietary fiber significantly increased the succinate-producing bacterial loads in the intestine, which promoted intestinal gluconeogenesis by binding fructose-1,6-bisphosphatase to relieve IR[25,26]. Our previous research has demonstrated that not all types of dietary fiber are effective in alleviating T2DM based on the modification of gut microbiota[27]. This highlights the need to screen bioactive dietary fibers that selectively regulate beneficial bacteria to alleviate IR.

In this study, we conducted a case-control study of Chinese adults with T2DM and found that the abundance of *P. distasonis* was inversely correlated with the severity of IR phenotypes. Hence, we screened for potential candidates that can promote the growth of *P. distasonis* from different dietary fibers and found that the bacteria increase significantly by *Dendrobium officinale* polysaccharides (DOP). *P. distasonis* plays a causal role in the alleviation of IR by promoting the production of nicotinic acid (NA) and regulating the NA-activated GPR109a pathway in the intestine, which is attributed to the improvement of intestinal barrier function. Another independent human T2DM cohort also demonstrated IR was inversely associated with the level of *P. distasonis* and NA. These results highlight the role of the DOP-*P. distasonis*-NA-GPR109a axis in improving IR.

## Results

### *P. distasonis* is negatively correlated with IR in patients with T2DM

To investigate the relationship between gut microbiota and IR in patients with T2DM (Supplementary Table 1), we first performed 16S rRNA gene sequencing of fecal samples to investigate gut microbiota changes in T2DM. Regarding the phylum differences, the T2DM group had higher Bacteroidetes and Proteobacteria than the health control (HC) group (Supplementary Fig. 1a). Detailed analysis of the top 25 bacterial genera illustrated that the T2DM group had a higher preponderance of *Prevotella*, unclassified *Enterobacteriaceae*, and *Megasphaera*, while that of *Parabacteroides*, *Faecalibacterium*, *Bacteroides*, *Roseburia*, unclassified *Ruminococcaceae*, unclassified *Lachnospiraceae*, *Oscillospira*, *Ruminococcus*, *Alistipes* and *Gemmiger* lower than HC group (Supplementary Fig. 1b). T2DM group was characterized by a significant decrease in taxa richness (Fig. 1a–c), and the composition of gut microbiota was different from the HC group ($p = 0.0010$, Fig. 1d). Based on the linear discriminant analysis (LDA) effect size (LEfSe) method, we found that the HC group was characterized by enriched *Faecalibacterium*, *Oscillospira*, *Parabacteroides*, *Dorea*, *Butyricimonas*, and *Bilophila*, while *Prevotella*, *Megasphaera*, and *Treponema* were dominant bacterial in the T2DM group (Fig. 1e, f). *Parabacteroides*, *Faecalibacterium*, *Bacteroides*, and unclassified *Lachnospiraceae* were positively correlated with the homeostasis model

assessment of β-cell function (HOMA-β) but negatively correlated with the homeostasis model assessment measure of insulin resistance (HOMA-IR), insulin, fasting blood glucose (FBG), and glycated hemoglobin (HbA1c) levels (Fig. 1g). *Parabacteroides* and *Faecalibacterium* were strongly correlated with most genera by network analysis (Supplementary Fig. 1c). Furthermore, we determined the abundance of representative strains belonging to *Parabacteroides* and *Faecalibacterium* and found that *P. distasonis*, *P. merdae*, *P. johnsonii*, and *F. prausnitzii* were significantly lower in the T2DM group compared to the HC group (Fig. 1h, i). Also, *P. distasonis* were most significantly correlated with most parameters related to IR (Fig. 1j). The above results suggest that *P. distasonis* has the potential to serve as a strain that indicates the severity of IR phenotypes.

### Enrichment of *P. distasonis* by bioactive dietary fiber

Bioactive dietary fiber alleviates IR and positively affects the host gut microbiota[24,28]. Supplementation with dietary fiber rich in DOP, inulin (IN), and β-glucan (BG) may enhance the abundance of *Parabacteroides* in the intestine[29–31]. We conducted an in vitro experiment to screen for bioactive dietary fiber that promotes the growth of *P. distasonis* (Fig. 2a), and found fecal bacteria are capable of effectively degrading DOP, IN, and BG, along with the significantly altered gut microbiota composition after in vitro fermentation (Fig. 2b, c, Supplementary Fig. 2a, b, f). At the genus level, DOP increased the abundance of *Parabacteroides* (39.73 times, Fig. 2b). There were no significant differences in α diversity between different groups (Supplementary Fig. 2c–e). However, the DOP group was characterized by the genera *Parabacteroides* and the species *P. distasonis*, which was further validated by qPCR (Fig. 2d–g). Furthermore, the IN group was characterized by the genera *Bifidobacterium* and *Candidatus Arthromitus* and the species *Bifidobacterium bifidum*, and the BG group was characterized by the genus *Prevotella* and the species *P. distasonis* (Supplementary Fig. 2g, h). Through network analysis, it was found that *Parabacteroides* were related to most genera (Supplementary Fig. 2i). Moreover, we also found a significant increase of *P. distasonis* in mice after the gavage with DOP (Fig. 2h, i). The above results suggest that DOP can significantly increase the abundance of *P. distasonis* in vivo and vitro.

### DOP alleviates HFD-induced IR

Supplementation with DOP did not affect the daily food intake in mice, but significantly reduced body weight, liver weight, epididymal fat weight, and epididymal fat weight to body weight ratio (Fig. 3a–c, Supplementary Fig. 3a–d). DOP also effectively improved dyslipidemia by reducing serum free fatty acids (FFA), triglyceride (TG), and total cholesterol (TC) (Supplementary Fig. 3e–g). Consistent with the improvement of lipid profiles, DOP treatment significantly reduced macrosteatosis, ballooning of hepatocytes, and intrahepatic triglyceride (IHTG) deposition in the liver of IR mice (Supplementary Fig. 3h). During oral glucose tolerance testing (OGTT) and insulin tolerance testing (ITT), DOP supplementation displayed significantly improved glucose tolerance and insulin sensitivity in IR mice (Fig. 3d–g). The fasting insulin level and HOMA-IR were also significantly lower in DOP-treated mice than in HFD-treated mice (Fig. 3h, i).

Inflammation contributes to the development of HFD-induced IR[10,32]. As expected, DOP reduced serum TNF-α and IL-1β levels compared to the HFD group (Fig. 3j, k). Inflammation in the liver also contributes to systemic IR[33]. We found that macrophages accumulated in 'crown-like structures', a histological hallmark of inflammation, and DOP-treated mice showed little macrophage accumulation (Supplementary Fig. 3h). Moreover, our data showed that DOP significantly reduced the mRNA expression of TNF-α, IL-1β, and IL-6 in the liver of IR mice (Supplementary Fig. 3i). Notably, DOP significantly reduced serum lipopolysaccharide (LPS) levels and increased the expression of

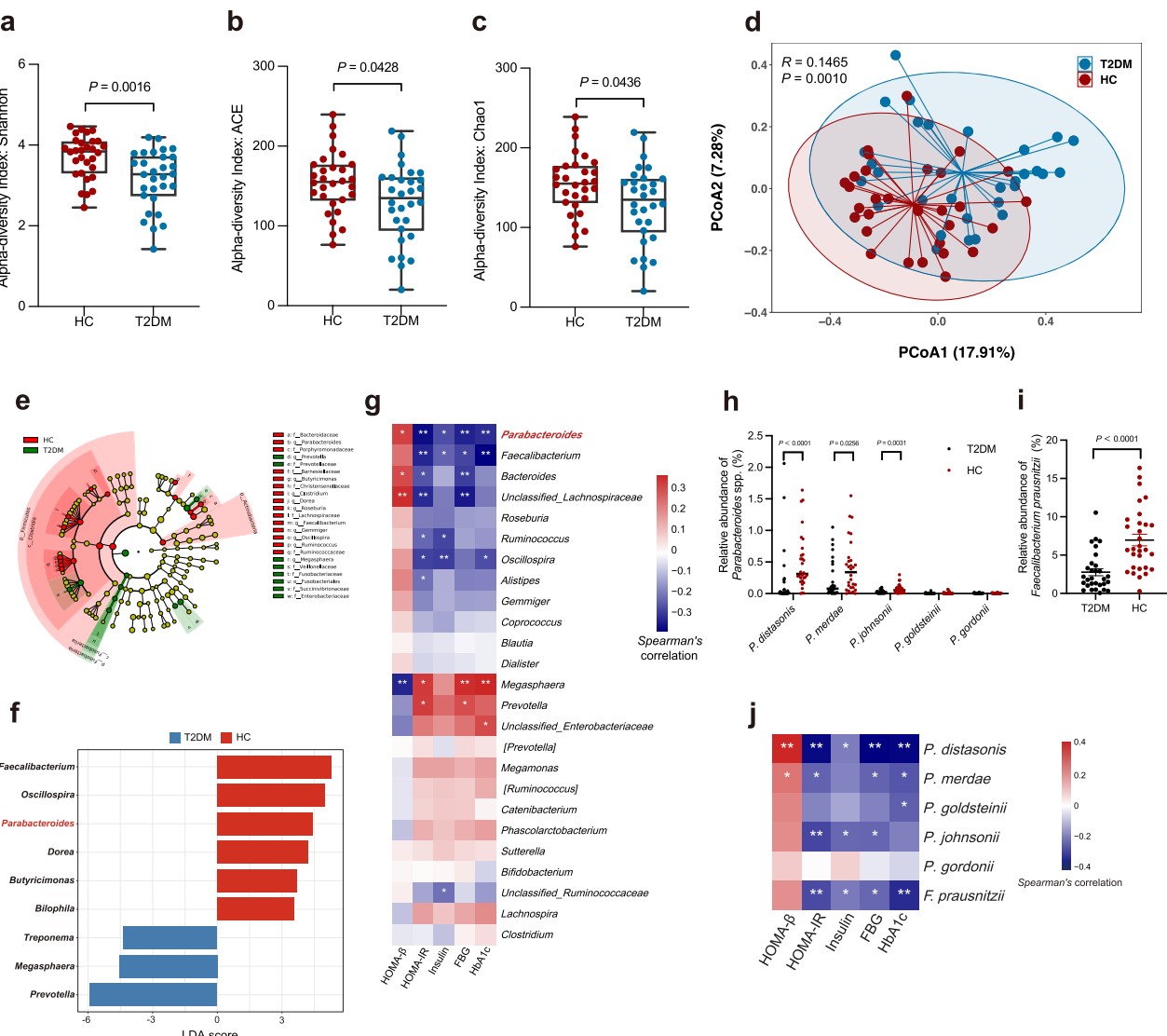

**Fig. 1 | Decreased abundance of *P. distasonis* is associated with IR.** The 16S rRNA gene amplicon sequencing datasets displayed the change of fecal microbiota in T2DM (*n* = 30) and HC (*n* = 30) groups. **a–c** α-diversity indicated by the Shannon, ACE, and Chao1 index. The box plots showing the minima, maxima, center, bounds of box and whiskers and percentile. **d** Principal coordinate analysis (PCoA) of all samples by weighted UniFrac distance (ANOSIM test). **e** Taxonomic cladogram generated from LEfSe analysis of 16S rRNA gene sequences. Each circle's size is proportional to the taxon's abundance. **f** LDA score representing the taxonomic data with significant differences between T2DM and HC groups. Only LDA scores >3 are shown. Blue indicates enriched taxa in the T2DM group. Red indicates enriched taxa in the HC group. **g** Spearman correlations (two-tailed Spearman's rank test) between the genus-level abundance and IR phenotypes. The color

represents positive (red) or negative (blue) correlations and FDRs are denoted: *FDR < 0.05; **FDR < 0.01. **h** The relative abundance of fecal *P. distasonis, P. merdae, P. johnsonii,* and *P. gordonii* was determined by qPCR. Data are presented as the mean ± SEM. *n* = 30 per group. **i** The relative abundance of fecal *F. prausnitzii* was determined by qPCR. Data are presented as the mean ± SEM. *n* = 30 per group. **j** Spearman correlations (two-tailed Spearman's rank test) between the species-level abundance and IR phenotypes. The color represents positive (red) or negative (blue) correlations and FDRs are denoted: *, FDR < 0.05; **, FDR < 0.01. T2DM (type 2 diabetes mellitus); HC (health control). Statistical analysis was performed using two-tailed Mann–Whitney test for **a–c**, **h**, and **i**. Source data are provided as a Source Data file.

Claudin-1, Occludin, Muc-2, and ZO-1 in the intestine of IR mice (Fig. 3l–n). These results collectively show that DOP produces anti-obesogenic, anti-insulin resistance, and anti-inflammatory effects and improves intestinal permeability in IR mice.

Dietary fiber significantly affects the gut microbiota and decelerates the development of IR[23]. We found no significant differences in the gut microbiota between the DOP and HFD groups at the phylum level (Supplementary Fig. 3j). However, DOP increased the abundance of *Parabacteroides*, *[Ruminococcus]*, and *Dorea* at the genus level (Supplementary Fig. 3k). There was no significant difference between the two groups in α-diversity, whereas a noticeable change in the composition of the gut microbiota after DOP treatment (Supplementary Fig. 3l–o). Consistent with the results of fiber screening, DOP also

significantly increased the abundance of *P. distasonis* (Fig. 3o, p, Supplementary Fig. 3p). Furthermore, *Parabacteroides* were negatively correlated with body weight gain, insulin, HOMA-IR, AUC of OGTT, LPS, TNF-α, and IL-1β, and *P. distasonis* showed a significant negative correlation with body weight gain, HOMA-IR, LPS, and TNF-α (Fig. 3q, Supplementary Fig. 3q).

**Alleviation of IR by DOP is microbiota-dependent**

The dysbiosis of the gut microbiota is intricately associated with the development and progression of IR[12], and the data above show that DOP can significantly alter the composition of the gut microbiota of IR mice. To investigate whether the change in the gut microbiota by DOP supplementation was involved in the alleviation of IR,

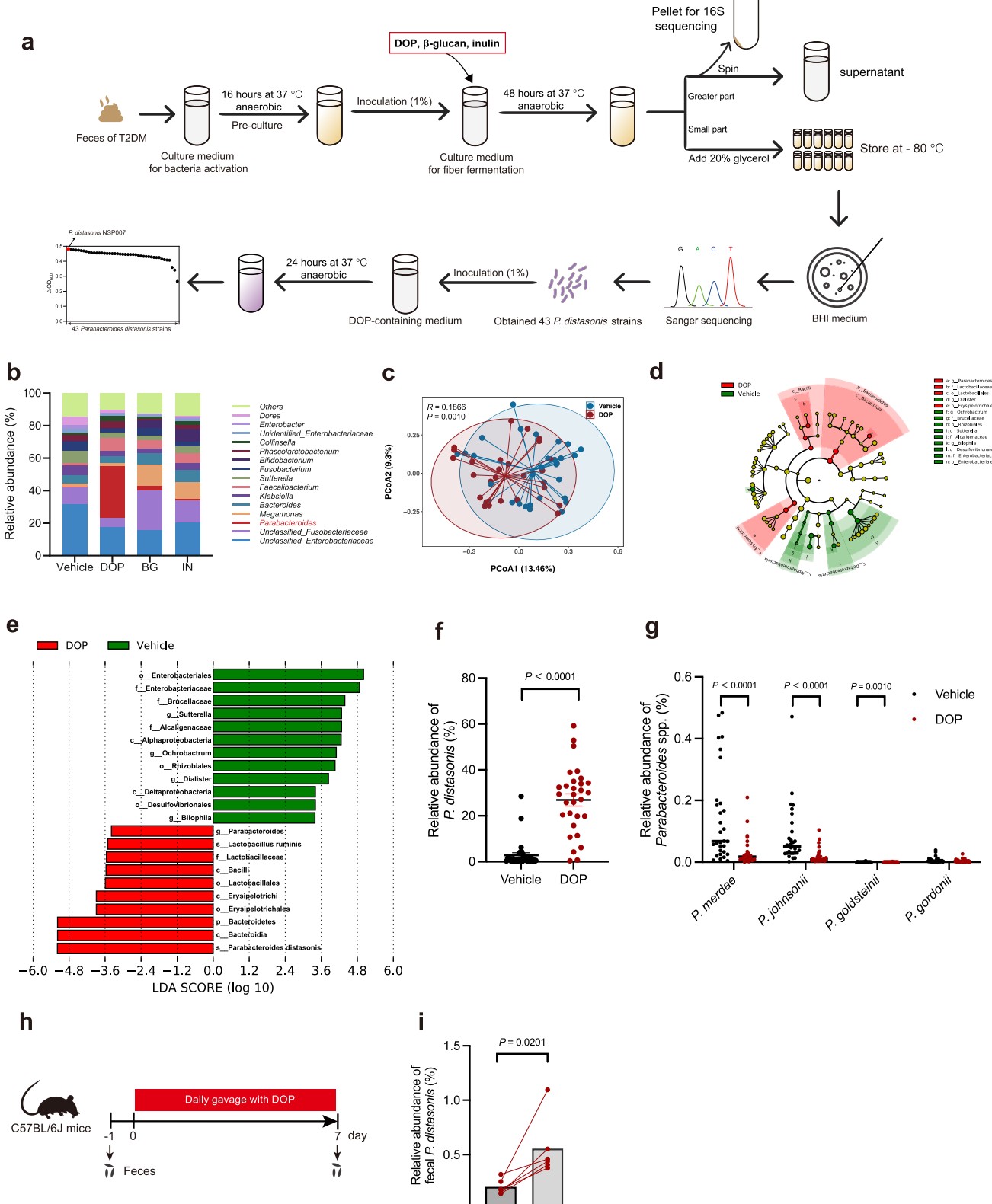

antibiotic-treated mice were fed HFD and administered DOP (Supplementary Fig. 4a). It was found that there were no significant differences in body weight change, food intake, OGTT, ITT, insulin, HOMA-IR, LPS, FFA, liver weight, epididymal fat weight, and epididymal fat weight to body weight ratio between Abx + vehicle and Abx + DOP group (Supplementary Fig. 4b–o). This suggests that the alleviation in DOP of IR depends on the gut microbiota.

To investigate the effect of DOP-influenced microbiota on IR mice, we transferred the microbiota from DOP-treated IR mice to conventional recipient IR mice (Supplementary Fig. 5a). DOP-FMT had no effect on food intake but significantly reduced body weight change, liver weight, and epididymal fat weight to body weight ratio (Supplementary Fig. 5b–g). DOP-FMT effectively improved dyslipidemia in IR mice by reducing the concentrations of TC (Supplementary Fig. 5h–j).

**Fig. 2 | *P. distasonis* is selectively enriched by DOP. a** Schematic of *P. distasonis* enrichment by DOP in vitro. Briefly, we incubated these three dietary fibers (DOP, or inulin or β-glucan) independently with fecal bacteria from T2DM patients for 48 h. The majority of the fermented broth underwent centrifugation, while the resulting precipitation was used for 16S rRNA gene sequencing. Simultaneously, the supernatant was employed for the sugar content determination. Another portion of the fermented broth was coated in BHI agar medium, resulting in the isolation of 43 strains of *P. distasonis*. Finally, the 43 strains of *P. distasonis* were cultured in BHI medium (DOP as the sole carbon) for 24 h, and the $OD_{600}$ was measured. The strain with the highest $\triangle OD_{600}$ was selected for further experiments. The tube in the image is referenced from Figure 1c of Ziesack et al (https://doi.org/10.1128/msystems.00352-19) with appropriate color modifications. **b** Relative abundance of bacteria at the genus level in different groups. **c** Principal coordinate analysis (PCoA) of DOP and Vehicle group by weighted UniFrac distance (ANOSIM test). **d** Taxonomic cladogram generated from LEfSe analysis (DOP and Vehicle group) by 16S rRNA gene amplicon sequencing. Each circle's size is proportional to the taxon's abundance. **e** LDA score representing the taxonomic data with significant differences between DOP and Vehicle groups. Only LDA scores >2 are shown. Green indicates enriched taxa in the Vehicle group. Red indicates enriched taxa in the DOP group. **f** The relative abundance of *P. distasonis* in DOP and Vehicle group was determined by qPCR. $n = 30$ per group. **g** The relative abundance of *P. merdae, P. johnsonii, P. goldsteinii,* and *P. gordonii* in DOP and Vehicle groups were determined by qPCR. $n = 30$ per group. **h–i** Mice were treated with 400 mg/kg DOP by daily gavage for 7 days. **h** Experimental scheme for i. $n = 6$ mice per group. **i** The change of relative abundance of fecal *P. distasonis* by DOP treatment. DOP (*Dendrobium officinale* polysaccharide); BG (β-glucan); IN (inulin). Data are presented as the mean ± SEM. Statistical analysis was performed using two-tailed Mann–Whitney test for **f** and **g**, two-tailed paired *t*-test for **i**. Source data are provided as a Source Data file.

Importantly, levels of OGTT, ITT, insulin, HOMA-IR, IL-1β, TNF-α, and LPS were significantly lower in the DOP-FMT group compared with the HFD-FMT group (Supplementary Fig. 5k–s), and DOP-FMT significantly increased the mRNA expression of *Claudin1, Muc2, Occludin,* and *Zo1* in the intestine of IR mice (Supplementary Fig. 5t). Moreover, the abundance of *P. distasonis* was significantly higher in the feces of the DOP-FMT group compared to the HFD-FMT group (Supplementary Fig. 5u). This data cumulatively provides cohesive evidence that DOP-influenced microbiota is responsible for alleviating HFD-induced IR.

## *P. distasonis* alleviates HFD-induced IR

To investigate the effect of *P. distasonis* on IR mice, we conducted the IR mice, and treated with *P. distasonis* NSP007 for five weeks (Fig. 4a). Oral administration of LPD substantially enhanced the abundance of *P. distasonis* in the feces of mice (Fig. 4b). LPD group showed a significant reduction in body weight change, liver weight, epididymal fat weight, and epididymal fat weight to body weight ratio compared to the Vehicle and KPD group (Fig. 4c, d, Supplementary Fig. 6a–d). LPD effectively improved dyslipidemia in IR mice by reducing the concentrations of FFA, TG, and TC (Supplementary Fig. 6e–g), along with significantly reduced macrosteatosis, hepatocyte ballooning, and IHTG deposition in the liver of IR mice (Supplementary Fig. 6h). A significant effect of LPD on IR mice is the improvement of OGTT, ITT, insulin, as well as HOMA-IR (Fig. 4e–j). Our histological analysis of the liver of IR mice found that LPD reduced the number of 'crown-like structures' (Supplementary Fig. 6h), and *P. distasonis* also reduced the mRNA expression of TNF-α and IL-1β in the liver of IR mice (Supplementary Fig. 6i). Furthermore, LPD significantly reduced the serum levels of TNF-α, IL-1β, LPS, and increased the expression of intestinal Claudin-1, Muc-2, Occludin, and ZO-1 in IR mice (Fig. 4k–o). These results collectively show that LPD treatment effectively reversed IR and metabolic disorders in HFD-induced IR mice.

## *P. distasonis*-derived nicotinic acid is negatively correlated with IR

Metabolites derived from the gut microbiota play an essential role in mediating the intricate interactions between the gut microbiota and the host[15]. Our results found that only living *P. distasonis* but not heat-killed *P. distasonis* can alleviate intestinal barrier function and IR in mice (Fig. 4e–j, n, o), implying that the beneficial effects of *P. distasonis* may rely on the production of bioactive metabolites. We performed non-targeted LC-MS metabolomics to identify the *P. distasonis* NSP007-derived metabolites in vitro. The metabolic profile of *P. distasonis* NSP007 showed a clear differentiation compared to the Vehicle group (Fig. 5a). The influenced metabolic pathway by *P. distasonis* NSP007 includes the metabolism of cofactors and vitamins, carbohydrates, nucleotide, and amino acids (Fig. 5b). A major influence of *P. distasonis* NSP007 was observed in the nicotinate and nicotinamide metabolism belonging to cofactors and vitamins pathway (Fig. 5b, c).

The heat map analysis confirmed that *P. distasonis* NSP007 significantly increased nicotinic acid (NA) and nicotinamide (NAM) concentrations compared to the Vehicle group (Fig. 5d). The identity of NA (Fig. 5e) and NAM (Supplementary Fig. 7a) was also confirmed by comparing the MS/MS fragmentation spectra of standard and experiment samples. Furthermore, the ability of *P. distasonis* NSP007 to produce NA (Fig. 5f, g) and NAM (Supplementary Fig. 7b, c) was verified by LC-MS/MS. Based on a series of methodology studies for NA detection, we found the limits of detection (LOD) is 0.23 ng/mL, recovery of NA was in the range of 97.4–103.1% at three spiked levels, with relative standard deviations (RSDs) of 2.5–4.3% (Supplementary Table 2). Analysis of the Kyoto Encyclopedia of Genes and Genomes (KEGG) pathway showed that the encoded genes of the *P. distasonis* NSP007 genome are mainly involved in membrane transport, signal transduction and metabolism of carbohydrates, amino acids, and cofactors and vitamins (including NA and NAM metabolism) (Fig. 5h).

To further investigate the relationship between *P. distasonis* and NA/NAM metabolism, we administered *P. distasonis* NSP007 to mice by gavage daily for seven days (Fig. 5i). The results showed that treatment with *P. distasonis* NSP007 significantly increased the NA level in mouse feces (Fig. 5j, k) but had no noticeable effect on the content of NAM (Supplementary Fig. 7d). Furthermore, we verified that all 43 isolated *P. distasonis* strains, as well as the commercial *P. distasonis* ATCC8503, were capable of producing NA (Supplementary Fig. 7e). In addition, we assessed NA and NAM levels in colonic contents in previously DOP-treated (Fig. 3a) and LPD-treated mice (Fig. 4a). The accumulation of NA was consistently observed in the LPD and DOP groups compared to the Vehicle group (Fig. 5l, m), with no difference observed in NAM (Supplementary Fig. 7f, g). We hypothesized that the level of NA in the colonic contents was correlated with the abundance of *P. distasonis*. Consistent with this notion, the abundance of *P. distasonis* was positively correlated with the level of NA in the colonic contents of mice (Fig. 5n). Furthermore, we found that the fecal NA level of the T2DM was significantly lower than HC group, but there was no difference in NAM level (Fig. 5o, Supplementary Fig. 7h). Moreover, the level of NA was negatively correlated with body weight gain, HOMA-IR, LPS, TNF-α, and IL-1β in IR mice (Supplementary Fig. 7i). We also observed the similar correlation (such as *P. distasonis*, HOMA-β, HOMA-IR, insulin, FBG, and HbA1c) in human cohorts (Fig. 5p, Supplementary Fig. 7j). To establish the relative contribution of gut microbiota to NA in the host, mice were treated with a cocktail of antibiotics to deplete the microbiota. We found that antibiotics were sufficient to deplete NA in the mouse feces. In addition, the fecal NA level was restored following fecal transplantation from normal mice (Fig. 5q, r). We further assessed the ability of the reported gut bacteria to produce NA in vitro and in vivo[34]. The in vitro data showed that all seven bacteria strains were able to produce NA (Supplementary Fig. 7k). However, the level of NA was highest after colonization of *P. distasonis* NSP007 in mice (Supplementary Fig. 7l–n). These results

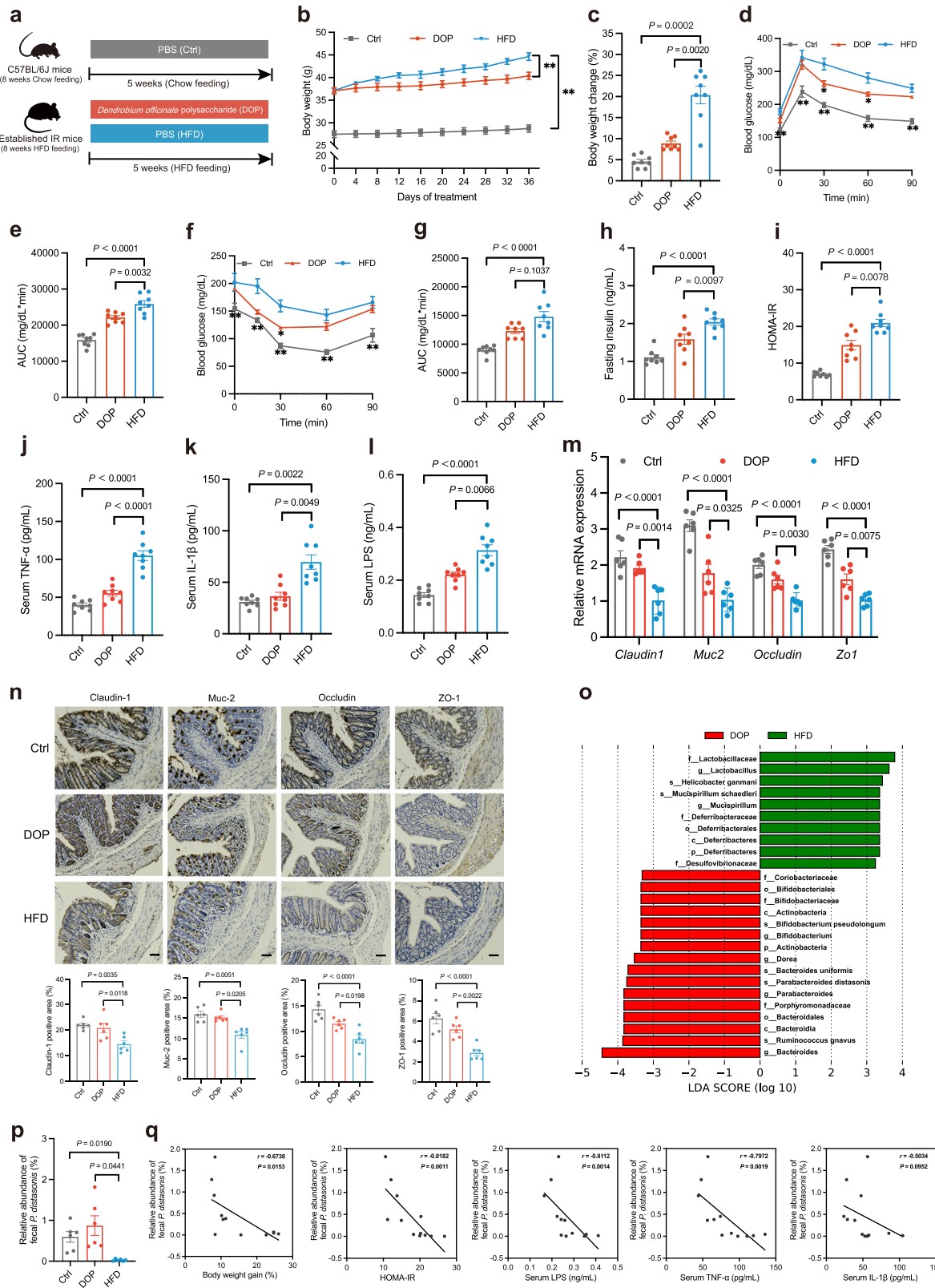

indicate that NA may be an important bioactive metabolite for the alleviation of IR by *P. distasonis*.

## Nicotinic acid and *P. distasonis* alleviate IR by activating GPR109a

Since bacteria derived-metabolites could have a direct impact on intestinal epithelial cell function[35], we evaluated the functional activity

of NA in vitro. The results showed that NA could significantly promote the mRNA expression of G-protein-coupled receptor 109a (GPR109a) in the intestinal epithelial cell line, Caco-2 (Fig. 6a). Furthermore, NA significantly increased the mRNA expression of *Claudin1*, *Muc2*, *Occludin*, and *Zo1* in Caco-2 cells (Fig. 6b). Because NA has been reported as a ligand for GPR109a[36], we evaluated whether NA mediated the expression of genes related to intestinal permeability through

**Fig. 3 | DOP enrich *P. distasonis* and ameliorates HFD-induced IR.** After an 8-week HFD treatment, mice were given PBS (HFD group) or DOP (DOP group) for 5 weeks, and the control group (Ctrl) was fed with a chow diet and given an equivalent volume of PBS. **a** Experimental scheme for (**b**–**p**). **b**–**l** *n* = 8 mice per group, **m**–**p** *n* = 6 mice per group. **b** Body weight curve. **c** Body weight change (%). OGTT (**d**) and AUC (**e**). ITT (**f**) and AUC (**g**). **h** Fasting insulin level. **i** HOMA-IR index. **j** Serum TNF-α level. **k** Serum IL-1β level. **l** Serum LPS level. **m** Relative mRNA expression of genes related to intestinal permeability. **n** Claudin-1, Muc-2, Occludin, and ZO-1 immunohistologic staining of colonic sections (top), and quantified positive area (bottom). Scale bars, 50 μm. **o** LDA score represents the taxonomic data with significant differences between DOP and HFD group. Only LDA scores >3 are shown. Green indicates enriched taxa in the HFD group. Red indicates enriched taxa in the DOP group. **p** The relative abundance of *P. distasonis* in different groups assessed by qPCR. **q** Spearman correlations (two-tailed Spearman's rank test) between the fecal *P. distasonis* abundance and IR phenotypes. DOP (*Dendrobium officinale* polysaccharide); HFD (High-fat diet). Data are presented as the mean ± SEM. Statistical analysis was performed using One-way ANOVA with Tukey's post hoc test for **b**, 0, 15, 60, and 90 min of **d**, **e**, 60 and 90 min of **f**, **h**, **i**, **j**, **m**, Claudin-1, Occludin, and Zo-1 of **n**, One-way ANOVA with Dunnett's T3 post hoc test for **c**, 30 min of **d**, 30 min of **f**, **g**, **k**, **l**, and **p**, Kruskal–Wallis test for 0 and 15 min of **f**, and Muc-2 of **n**. **P* < 0.05; ***P* < 0.01. Source data are provided as a Source Data file.

GPR109a. As shown in Fig. 6c, NA failed to induce the mRNA expression of *Claudin1*, *Muc2*, *Occludin*, and *Zo1* in Caco-2 cells pretreated with mepenzolate bromide (MPN, an inhibitor of GPR109a). Furthermore, the mRNA expression of *Gpr109a* was significantly higher in DOP, DOP-FMT, and LPD-treated mice compared to the Vehicle-treated mice (Fig. 6d–f). Based on the above results, we speculate that NA plays an important role in DOP and *P. distasonis* NSP007 in improving intestinal barrier function and IR, and through the activation of intestinal GPR109a.

To investigate the effect of NA on IR mice, mice were pretreated with HFD for eight weeks, then administrated with PBS, NA, MPN, and NA plus MPN for a further five weeks (Fig. 6g). The NA group showed a significant reduction in body weight, liver weight, and epididymal fat weight compared to the Vehicle, MPN, and NA + MPN groups (Fig. 6h, i, Supplementary Fig. 8a–d). NA improved dyslipidemia and liver injury in IR mice by reducing the levels of FFA, macrosteatosis, hepatocyte ballooning, and IHTG deposition in the liver (Supplementary Fig. 8e–i). Furthermore, NA significantly improved OGTT, ITT, insulin, and HOMA-IR compared to the Vehicle, MPN, and NA + MPN treatment (Fig. 6j–o). Moreover, NA significantly reduced the counts of 'crown-like structures' and the mRNA expression of TNF-α and IL-6 in the liver of IR mice (Supplementary Fig. 8h, i). With the improvement of intestinal barrier function by *P. distasonis*, we also found NA significantly reduced the serum levels of TNF-α, IL-1β, and LPS levels, and increased mRNA expression of intestinal *Claudin1*, *Muc2*, *Occludin*, and *Zo1* in IR mice, accompanied by the enhanced mRNA expression of *Gpr109a* (Fig. 6p–t). Furthermore, the protein expressions of Claudin-1, Occludin, Muc-2, and ZO-1 in the intestine were increased in NA-treated mice (Supplementary Fig. 8j). Taken together, these results demonstrated the protective effect of NA on HFD-induced IR in a GPR109a activation-dependent manner.

To investigate whether the beneficial effects of *P. distasonis* NSP007 in improving IR are dependent on the activation of GPR109a, mice were pretreated with HFD for eight weeks, then administrated with PBS, *P. distasonis* NSP007, MPN, and *P. distasonis* NSP007 plus MPN for a further five weeks (Supplementary Fig. 9a). The supplementation of MPN eliminated the improvement of *P. distasonis* NSP007 on weight gain, fat accumulation, IR, chronic inflammation, and intestinal permeability in IR mice (Supplementary Fig. 9b–t).

### Lower levels of *P. distasonis* and nicotinic acid in T2DM patients: validation cohort

To explore the relationship between *P. distasonis* and NA with the severity of IR phenotypes in humans, we enrolled another independent validation cohort (Supplementary Table 3). Compared to healthy individuals, the T2DM patients had lower levels of *P. distasonis* and NA (Fig. 7a–c). *P. distasonis* abundance and the NA level showed a significantly negative correlation with FBG, insulin, and HOMA-IR (Fig. 7d–i). Furthermore, patients with T2DM generally use anti-diabetic medications, we also investigate the effects of main anti-diabetic drugs (metformin, gliclazide, and glibenclamide) taken by T2DM patients on the growth of *P. distasonis* (Supplementary Tables 1 and 3). The results showed that these drugs had no significant effect on

the growth of *P. distasonis* NSP007 (Supplementary Fig. 10a–c). Furthermore, reanalysis of Human Microbiome Project (HMP) data demonstrated a negative connection between the abundance of *Parabacteroides* and body mass index (BMI), fasting plasma glucose (FPG), and steady-state plasma glucose (SSPG) (Fig. 7j–l). These data suggest the decreased abundance of *P. distasonis* and level of NA may be a vital factor in the pathogenesis of IR.

## Discussion

The dysbiosis of the gut microbiota plays a vital role in the development and progression of T2DM[19]. IR is a crucial feature of T2DM, and the role of gut microbiota on IR has not been well studied[1,37]. Several previous human studies have indicated that *Parabacteroides* abundance is negatively correlated with metabolic disease severity[38–40]. *P. distasonis* is a Gram-negative anaerobic bacterium known to colonize the gastrointestinal tract of various species[41]. Our human cohort studies showed that *P. distasonis* abundance was negatively correlated with IR. Numerous studies support that *P. distasonis* plays a beneficial role in the treatment of inflammatory arthritis[42], metabolic dysfunction[26], colorectal tumourigenesis[43], and acute pancreatitis[44], whereas the detailed molecular mechanism of *P. distasonis* on IR is still unclear. Prolonged consumption of a high-fat diet impairs intestinal barrier function, allowing bacterial LPS to leak into circulation and trigger chronic systemic inflammation, which is a critical factor in the development of IR[8,9]. We found supplementation with *P. distasonis* improved the expression of intestinal tight junction proteins, decreased circulating LPS and pro-inflammatory factors levels, and improved IR in mice. These findings suggest that *P. distasonis* may improve IR by enhancing gut barrier function.

Numerous small-molecule metabolites derived from the gut microbiota play critical roles in shuttling information to the host, which helps maintain host health[45]. Consequently, defining and characterizing microbial-derived metabolites can aid in complementing microbial functions and deciphering host-microbiota interplay in health and disease[15]. Our results showed that heat-killed *P. distasonis* could not alleviate IR, indicating that bioactive metabolites produced by *P. distasonis* play an essential role in protecting against IR. In the present study, *P. distasonis* was shown to produce NA both in vivo and in vitro, which to our knowledge had not previously been reported. Additionally, we observed a significant decrease in NA levels in mouse feces after bacterial clearance by antibiotics, suggesting a crucial contribution of the gut microbiota towards the homeostasis of NA in the host. NA is a water-soluble vitamin which is generally used to treat lipid disorders[46]. Its supplementation has been found to enhance intestinal barrier function in piglet[47]. Importantly, NA acts as an exogenous agonist of GPR109a[36], and a recent study demonstrated that GPR109a deficiency compromised intestinal barrier function in mice[48], suggesting the crucial role played by GPR109a in maintaining intestinal barrier integrity. Our results indicated NA treatment significantly increased the mRNA expression of tight junction proteins and mucins in intestinal epithelial cells. Furthermore, NA treatment improved intestinal barrier function, and attenuated inflammation in HFD-induced IR mice and relies on GPR109a activation. As a result, we

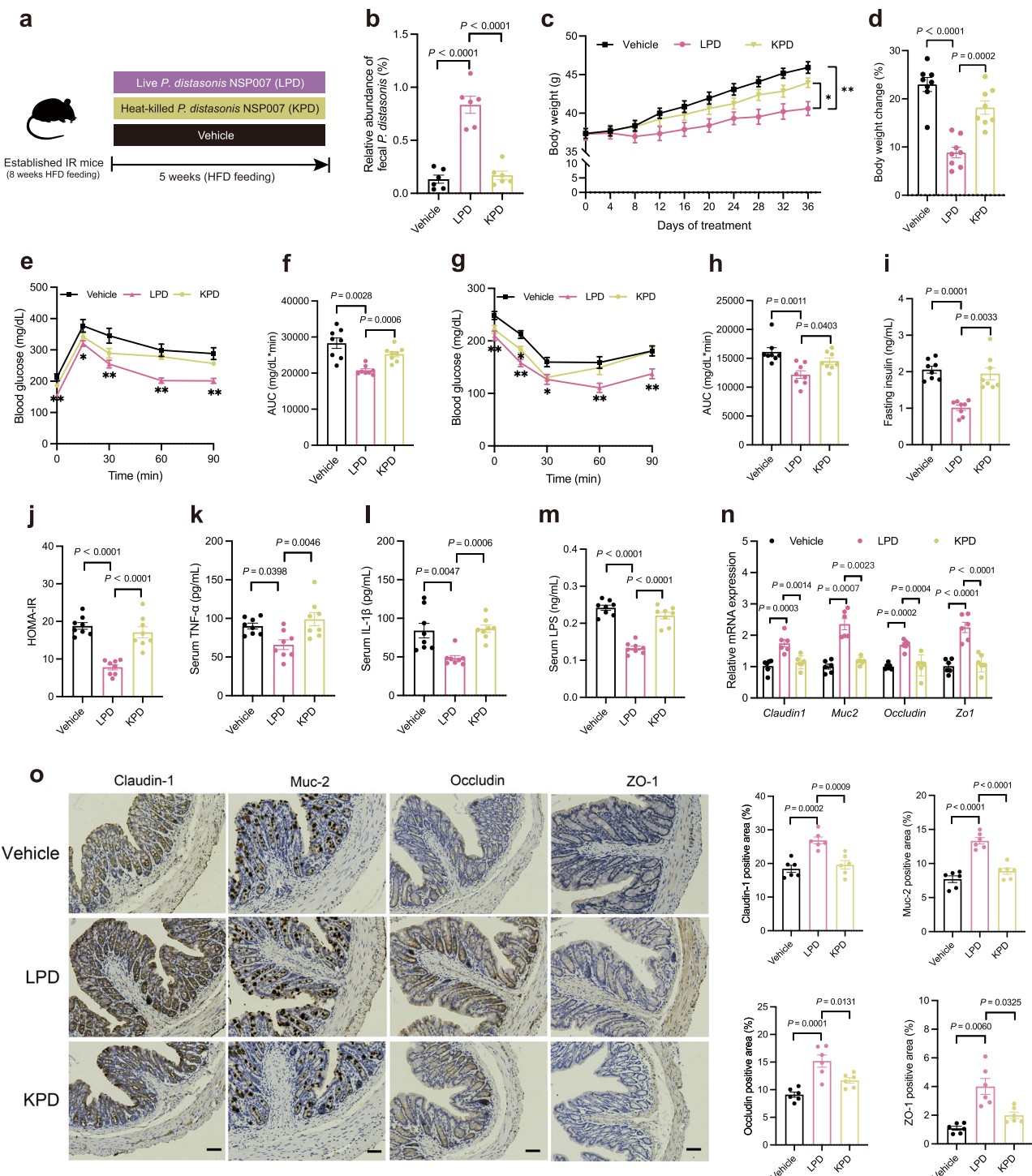

**Fig. 4 | *P. distasonis* ameliorates HFD-induced IR.** After an 8-week HFD treatment, mice were given PBS (Vehicle group), *P. distasonis* NSP007 (LPD group), or heat-killed *P. distasonis* NSP007 (KPD group) for 5 weeks. **a** Experimental scheme for **b**–**o**. **b**–**m** *n* = 8 mice per group, **n**–**o** *n* = 6 mice per group. **b** The relative abundance of *P. distasonis* in different groups assessed by qPCR. **c** Body weight curve. **d** Body weight change (%). OGTT (**e**) and AUC (**f**). ITT (**g**) and AUC (**h**). **i** Fasting insulin level. **j** HOMA-IR index. **k** Serum TNF-α level. **l** Serum IL-1β level. **m** Serum LPS level. **n** Relative mRNA expression of genes related to intestinal permeability. **o** Claudin-1,

Muc-2, Occludin, and ZO-1 immunohistologic staining of colonic sections (left), and quantified positive area (right). Scale bars, 50 μm. Data are presented as the mean ± SEM. Statistical analysis was performed using One-way ANOVA with Tukey's post hoc test for **b**–**d**, 0 and 30 min of **e**, 0 and 30 min of **g**, **j**, **k**, **m**, *Claudin1*, *Occludin*, and *Zo1* of **n**, and **o**, One-way ANOVA with Dunnett's T3 post hoc test for 60 and 90 min of **e**, **f**, **i**, and *Muc2* of **n**. Kruskal–Wallis test for 15 min of **e**, 15, 60, and 90 min of **g**, **h**, and **l**. *$P < 0.05$; **$P < 0.01$. Source data are provided as a Source Data file.

concluded that NA produced by *P. distasonis* could affect intestinal permeability by activating GPR109a in the intestine. In addition, we conducted a cohort study and found a significant negative correlation between the levels of *P. distasonis* and NA with FBG, insulin, and

HOMA-IR, which further emphasized the importance of *P. distasonis* and NA in improving IR.

Dietary fiber escapes gastrointestinal digestion and is degraded by the gut microbiota into nutrients that are tractable to the host and

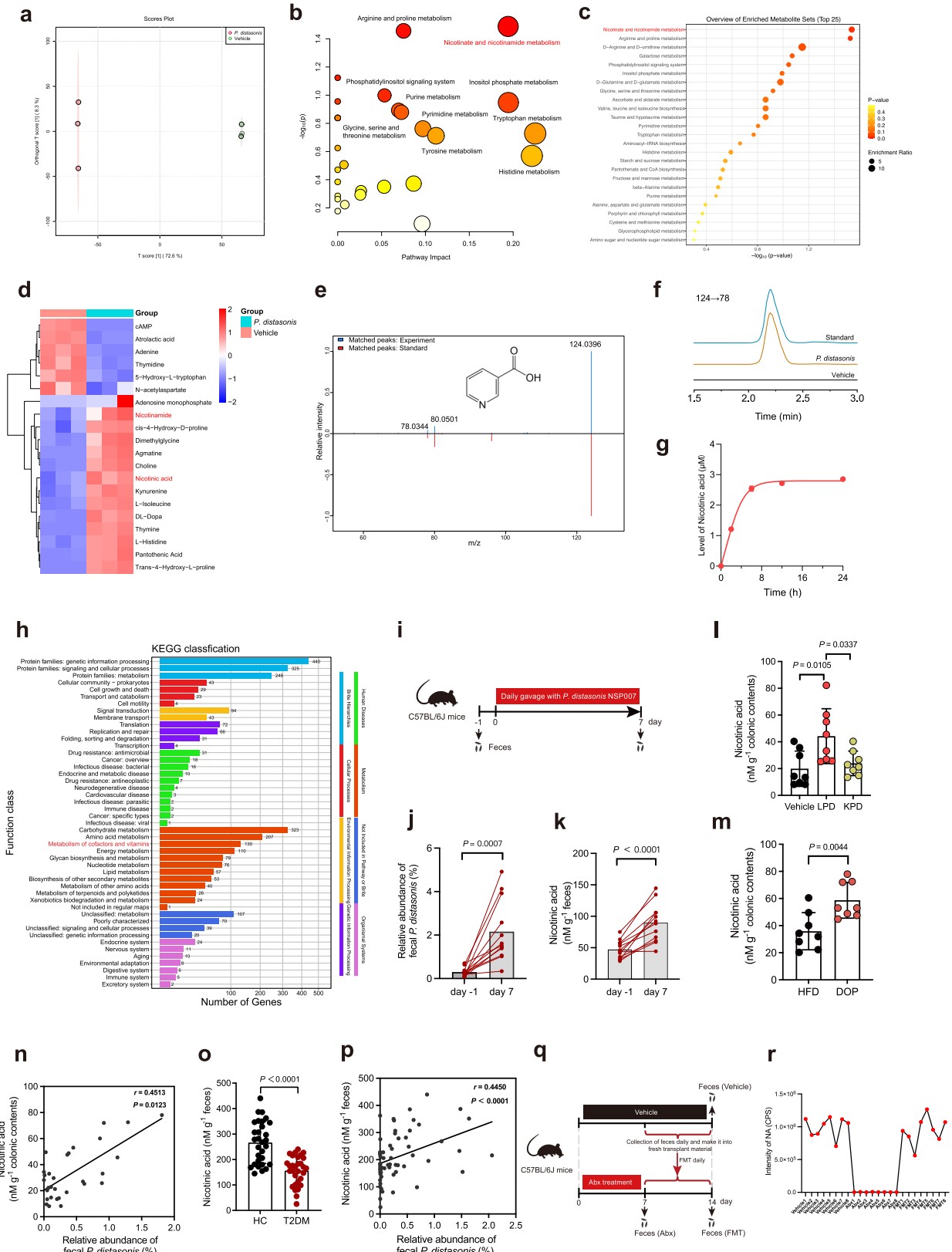

affect the composition of microbes[23]. Importantly, dietary fiber has been demonstrated to alleviate IR by promoting the growth of specific gut bacteria and facilitating the production of microbiota-derived metabolites[24]. Therefore, we hypothesized that selective promotion of *P. distasonis* in the gut by dietary fiber could slow down the progression of IR. In this study, we found that DOP significantly enriched the abundance of *P. distasonis* based on in vitro fermentation, and this finding was confirmed in DOP-treated mice. DOP has been reported to reduce obesity and change the gut microbiota composition; however, it is still unclear whether the gut microbiota modulated

**Fig. 5 | Nicotinic acid is a bioactive metabolite of *P. distasonis*. a–e** Non-targeted metabolomic analysis of the BHI medium inoculated with *P. distasonis* or not. $n = 3$ biologically independent samples per group. **a** OPLS-DA analysis of metabolic profile in the supernatant of *P. distasonis* and BHI medium. **b** The enriched pathways calculated from mummichog algorithm based on the nontargeted metabolomics analysis of samples from *P. distasonis* incubation (hypergeometric test). **c** Enrichment analysis of the top 25 enrichment metabolic pathways (hypergeometric test). **d** Heat map of metabolites involved in the significantly changed metabolic pathways. **e** The MS/MS spectra of NA standard and NA detected in *P. distasonis*-incubated samples. **f** Extracted ion chromatograms of NA from cultured *P. distasonis* compared to the Vehicle. **g** The intensity of NA in the culture supernatant of *P. distasonis* at different time points. $n = 3$ biologically independent samples per group. **h** The Kyoto Encyclopedia of Genes and Genomes (KEGG) pathway analysis of the encoded genes of *P. distasonis* NSP007. **i–k** Changes in the *P. distasonis* abundance and NA levels in the feces of mice before and after gavage with *P. distasonis* NSP007. Mice were treated with *P. distasonis* ($2 \times 10^8$ CFU/mice/d)

for 7 days. $n = 12$ mice per group. **i** Experimental scheme for **j–k**. **j** The relative abundance of *P. distasonis*. **k** The changes of fecal NA level. **l** Colonic NA level in Vehicle, LPD, and KPD group of mice. $n = 8$ mice per group. **m** Colonic NA level in HFD and DOP group of mice. $n = 8$ mice per group. **n** Spearman correlations (two-tailed Spearman's rank test) between the colonic NA level and abundance of fecal *P. distasonis* in mice (Vehicle, LPD, KPD, HFD, and DOP group). **o** Fecal NA level in HC and T2DM group of humans. **p** Spearman correlations (two-tailed Spearman's rank test) between the colonic NA level and fecal *P. distasonis* abundance (HC and T2DM group). **q, r** Level of fecal NA in mice after antibiotic cocktails (Abx) or fecal microbiota transplantation (FMT) treatment. $n = 8$ mice per group. **q** Experimental scheme for **r**. **r** Fecal NA level. DOP (*Dendrobium officinale* polysaccharide); HFD (High-fat diet); LPD (*P. distasonis* NSP007); KPD (heat-killed *P. distasonis* NSP007); T2DM (type 2 diabetes mellitus); HC (health control). Data are presented as the mean ± SEM. Statistical analysis was performed using One-way ANOVA with Tukey's post hoc test for **l**, two-tailed paired *t*-test for **j** and **k**, and two-tailed unpaired *t*-test for **m** and **o**. Source data are provided as a Source Data file.

by DOP is a crucial factor contributing to the amelioration of IR[29]. Our study reveals that the beneficial impact of DOP on improving IR was deleted in antibiotic-treated mice, yet transplanting the microbiota obtained from DOP-treated mice into recipient mice ameliorated HFD-induced IR. Additionally, we found DOP significantly improved the intestinal barrier function and decreased circulating LPS, and pro-inflammatory factors levels of IR mice, and this effect was found to be dependent on the presence of gut microbiota. These findings demonstrated the essential role of DOP-responsive bacteria and its effect on intestinal barrier function in mitigating IR. Our results revealed that *P. distasonis* was the most responsive strain to DOP and was associated with improved IR. Given the growing appreciation for the intricate interplay between microbiota and host, it is imperative to examine the distinct functions of individual bacterial species or strains in gut microbiota[49]. *P. distasonis* NSP007, which we selected from human feces, effectively improved intestinal barrier function and ameliorated HFD-induced IR, highlighting its potential as a therapeutic candidate for IR and related metabolic diseases.

In summary, our study revealed the decreased levels of *P. distasonis* and NA in T2DM patients are significantly linked to IR deterioration. Supplementation with *P. distasonis* effectively improved the intestinal barrier function and IR, and this effect depends on the activation of GPR109a receptor in the intestine via NA produced by *P. distasonis*. Furthermore, we found that the bioactive dietary fiber DOP exhibits a favorable ability to prevent the development of IR by promoting the growth of *P. distasonis*. Our study highlights the significance of the *P. distasonis*-NA-GPR109a axis in improving IR and suggests that DOP represents a promising dietary strategy in the alleviation of IR and related metabolic diseases.

## Methods

### Preparation of *Dendrobium officinale* polysaccharides and β-glucan

The extraction method for *Dendrobium officinale* polysaccharides (DOP) refers to our previous work[50]. The dried *Dendrobium officinale* stem was powdered, immersed in hexane (1:5, w/v), and occasionally stirred for 24 h to remove impurities. This process was repeated once. After drying, the powder immersed in ethanol (1:5, w/v) was stirred occasionally for 24 h. This procedure was repeated only once. After drying, the powder was extracted with distilled water (1:20, w/v) at 70 °C for 4 h, followed by centrifugation ($10,000 \times g$, 25 °C, 20 min). The residue was then discarded. Ethanol was added to the supernatant until the concentration was 75 % (v/v). The collected residues were redissolved in distilled water, freeze-dried, and then redissolved in distilled water (1:99, w/v). The solution was treated with α-amylase (3000 units/mL, 70 °C, 2 h), then repeatedly freeze-thawed (−80 to 25 °C) thrice to remove amylase. After dialyzing (8000–12000 Da), the supernatant was precipitated with ethanol (1:3, w/v) and centrifuged.

The collected residues were redissolved in distilled water and subjected to dialysis and lyophilization to obtain DOP. β-glucan was obtained from barley, as described in our previous work[51].

### Human subjects

For cohort 1, all participants were recruited in the First Affiliated Hospital of Nanchang University and Nanchang University (Nanchang, China) and provided informed consent. The primary inclusion criteria included 30 type 2 diabetes patients (aged 25–70 years; Definite diagnosis of type 2 diabetes; 6.5% <HbA1c < 10%) and 30 healthy controls (matched the type 2 diabetes group in age, sex, native place, and BMI). Participants were excluded if they had gastrointestinal diseases, malignant tumors, autoimmune disorders, infectious diseases, renal dysfunction (severe renal disease creatinine >3.0 mg/dL), a history of gastrointestinal surgery in the previous year, or were administered antibiotics for more than three days in the previous three months. For cohort 2 (verification cohort), all participants were recruited at the First Affiliated Hospital of Nanchang University and Nanchang University (Nanchang, China) and provided informed consent. The primary inclusion criteria included 30 patients with type 2 diabetes and 30 healthy controls. Identical inclusion and exclusion criteria were applied to the discovery phase cohort. The study involving human participants was approved by the Ethics Committee of the First Affiliated Hospital of Nanchang University, number IIT2022076. The trial was registered at the Chinese clinical trial registry, number ChiCTR2200065715. We have received written informed consent from all the participants. Additional details on the trial design and protocol are given in the Chinese clinical trial registry (https://www.chictr.org.cn/showproj.html?proj=185349, registration number ChiCTR2200065715).

For the participants, peripheral venous blood was drawn in the morning the day after admission. Participants were provided with a stool sampler and detailed illustrated instructions for sample collection. Fresh stool samples collected from each participant were immediately transported to the laboratory and immediately frozen at −80 °C.

### Animal experiments

Male C57BL/6J mice (five weeks old) were purchased from Charles River Laboratories (Beijing, China). Male mice are more susceptible to IR than females[52,53]. All mice were housed under specific pathogen-free conditions with a 12-h light-dark cycle and provided with standard chow and water ad libitum, with temperature kept at 21–24 °C, and humidity at 40–70%. Mice were acclimatized for one week before any treatment. All animal experiments were performed under the Guidelines for Care and Use of Laboratory Animals of the National Institutes of Health and were approved by the Experimental Animal Care and Use Committee of Nanchang University, number IACUC- 20221030002.

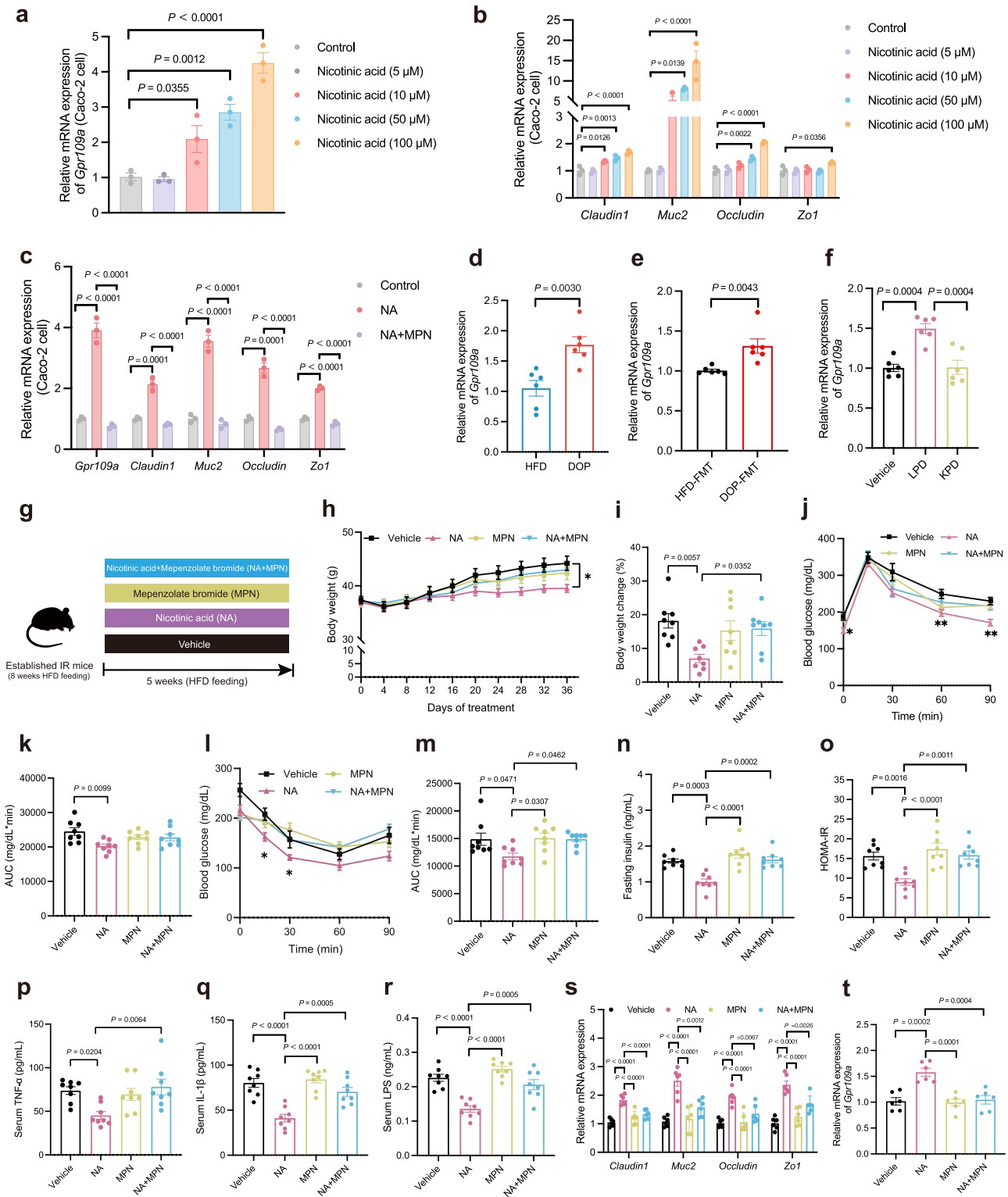

For the DOP efficiency assay in HFD-induced IR mice after an eight-week HFD (Research Diets, USA, Cat# D12492, the same below) treatment. Mice were given PBS or DOP (400 mg/kg/d) for 5 weeks, and the control group (Ctrl) was fed with a chow diet (Research Diets, USA, Cat# D12450J) all the time and given an equivalent volume of PBS (*n* = 8 mice in each group).

For the *P. distasonis* efficiency assay in HFD-induced IR mice, after an eight-week HFD treatment, mice were administered with *P. distasonis* NSP007 (2 × 10⁸ CFU/mice/d) or heat-killed *P. distasonis* NSP007 (2 × 10⁸ CFU/mice/d) for five weeks. The Vehicle group was

administered with an equivalent volume of PBS by gavage (*n* = 8 mice in each group).

To investigate the role of GPR109a in *P. distasonis* improving IR, after an eight-week HFD treatment, mice were administered with *P. distasonis* NSP007 (2 × 10⁸ CFU/mice/d), MPN (2 mg/kg/d), or *P. distasonis* NSP007 plus MPN for five weeks. The Vehicle group was administered with an equivalent volume of PBS by gavage (*n* = 8 mice in each group).

For the NA efficiency assay in HFD-induced IR mice, after an eight-week HFD treatment, mice were instilled with NA solution in PBS 1 mM

**Fig. 6 | NA improves intestinal integrity and IR. a, b** Caco-2 cells were treated with serum-free medium (Control) or medium added with NA for 12 h. *n* = 3 biologically independent samples per group. **a** *Gpr109a* expression in Caco-2 cells was examined using qPCR. **b** *Claudin1*, *Muc2*, *Occludin*, and *Zo1* expression in Caco-2 cells was examined using qPCR. **c** *Gpr109a*, *Claudin1*, *Muc2*, *Occludin*, and *Zo1* expression in Caco-2 cells was examined using qPCR. Caco-2 cells were pre-treated with serum-free medium (Control) or MPN for 1 h and subsequently stimulated with NA for another 12 h. *n* = 3 biologically independent samples per group. **d** *Gpr109a* expression in colonic tissues of mice in the HFD and DOP group. *n* = 6 mice per group. **e** *Gpr109a* expression in colonic tissues of mice in the HFD-FMT and DOP-FMT groups. *n* = 6 mice per group. **f** *Gpr109a* expression in colonic tissues of mice in the Vehicle, LPD, and KPD groups. *n* = 6 mice per group. **g–t** After an 8-week HFD treatment, mice were treated with PBS (Vehicle group), NA (NA group), MPN

(Mepenzolate bromide, MPN group), or a combination of NA and MPN (NA + MPN group) three times a week for 5 weeks. **h–r** *n* = 8 mice per group, **s–t** *n* = 6 mice per group. **g** Experimental scheme for **g–t**. **h** Body weight curve. **i** Body weight change (%). **j** OGTT and **k** AUC. **l** ITT and **m** AUC. **n** Fasting insulin level. **o** HOMA-IR index. **p** Serum TNF-α level. **q** Serum IL-1β level. **r** Serum LPS level. **s** Relative mRNA expression of genes related to intestinal permeability. **t** Relative mRNA expression of *Gpr109a* in different groups. DOP (*Dendrobium officinale* polysaccharide); HFD (High-fat diet); LPD (*P. distasonis* NSP007); KPD (Heat-killed *P. distasonis* NSP007). Data are presented as the mean ± SEM. Statistical analysis was performed using One-way ANOVA with Tukey's post hoc test for **a–c**, **f**, **h–k**, 15 min of **l–o**, **q–t**, One-way ANOVA with Dunnett's T3 post hoc test for **p**, two-tailed unpaired *t*-test for **d**, two-tailed Kruskal–Wallis test for **e** and 30 min of **l**. *\*P* < 0.05; *\*\*P* < 0.01. Source data are provided as a Source Data file.

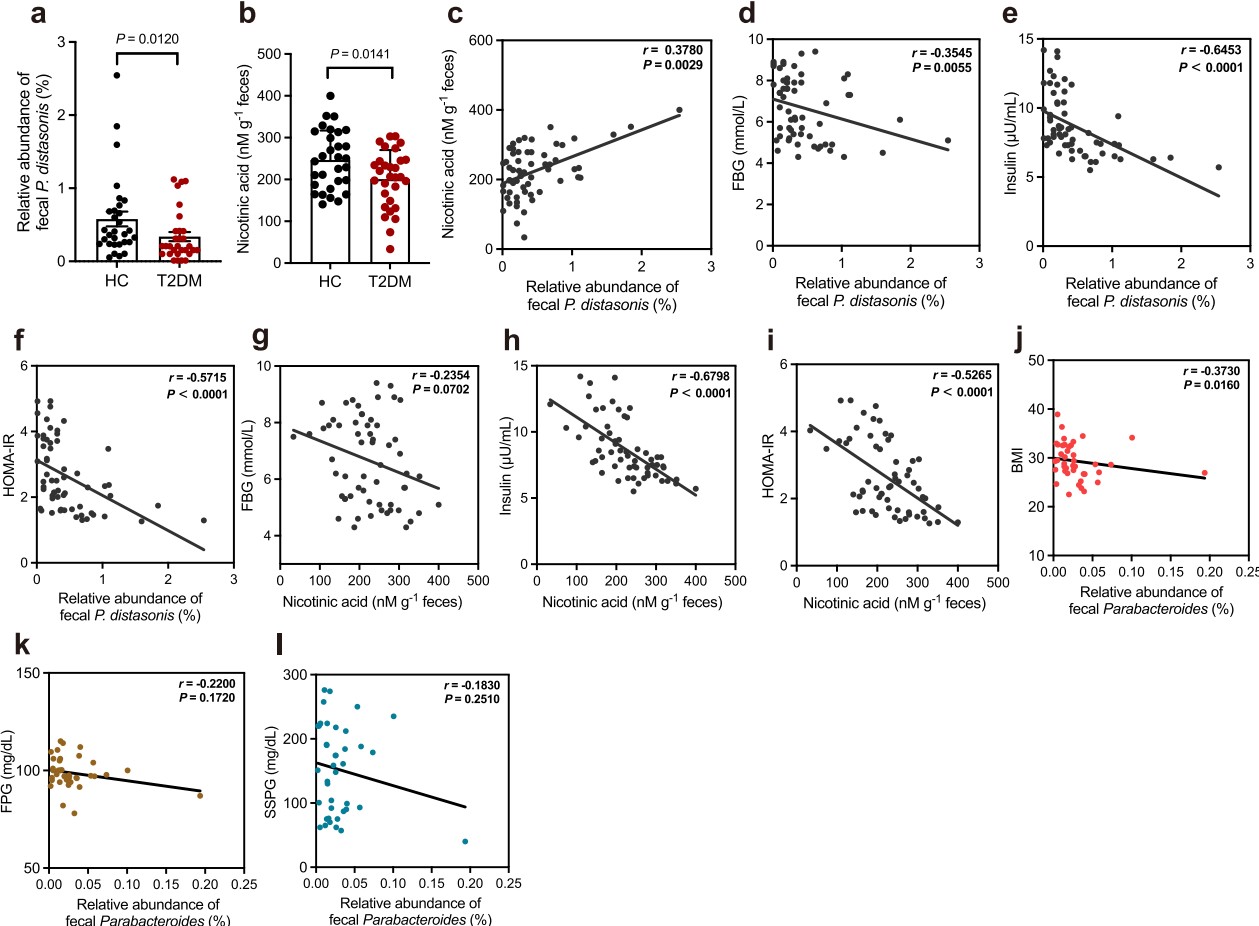

**Fig. 7 | *P. distasonis* abundance and fecal NA levels are negatively correlated with IR in human. a** The relative abundance of *P. distasonis* in HC (*n* = 30) and T2DM (*n* = 30) groups. **b** The level of fecal NA in HC (*n* = 30) and T2DM (*n* = 30) groups. **c** Spearman correlations (two-tailed Spearman's rank test) between the fecal NA level and *P. distasonis* abundance in the validation cohort. Spearman correlations (two-tailed Spearman's rank test) between the *P. distasonis* abundance and (**d**) FBG (fasting blood glucose) level, (**e**) insulin level, and (**f**) HOMA-IR (homeostasis model assessment measure of insulin resistance) index in the validation cohort. Spearman correlations (two-tailed Spearman's rank test) between

the fecal NA level and (**g**) FBG, (**h**) insulin level, and (**i**) HOMA-IR index in the validation cohort. Spearman correlations (two-tailed Spearman's rank test) between the abundances of *Parabacteroides* and (**j**) BMI (body mass index), (**k**) FPG (Fasting plasma glucose) level, and (**l**) SSPG (steady-state plasma glucose) level. The data is from the Human Microbiome Project (https://www.hmpdacc.org/hmp/). T2DM (type 2 diabetes mellitus); HC (health control). Data are presented as the mean ± SEM. Statistical analysis was performed using two-tailed Mann–Whitney test for **a** and unpaired *t*-test for **b**. Source data are provided as a Source Data file.

(200 μL), MPN solution in PBS 10 μM (200 μL), or NA combined with MPN by intrarectal route three times a week for five weeks and treated with HFD. The Vehicle group was administered with an equivalent volume of PBS by gavage (*n* = 8 mice in each group).

For intestinal microbiota depletion, mice were treated with antibiotic cocktails (ampicillin, 100 mg/kg; metronidazole, 100 mg/kg;

vancomycin, 50 mg/kg; neomycin, 100 mg/kg) by daily gavage for seven consecutive days (*n* = 8 mice in each group)[54].

For fecal microbiota transplantation, the microbiota donor mice were treated with 400 mg/kg/day of DOP or PBS for two weeks, followed by a daily collection of feces. Feces from each group were pooled and transferred to an anaerobic chamber. Approximately

100 mg of feces were resuspended in 1 mL of sterile anaerobic PBS, shaken for 3 min, and centrifuged ($500 \times g$, 4 °C for 1 min), the settled suspension was administered by oral gavage to recipient mice. After eight weeks of HFD treatment, recipient mice were pretreated with an antibiotic cocktail as mentioned above, followed by a daily oral gavage of 200 μL suspension for five weeks ($n = 8$ mice in each group).

For the effect of DOP on the fecal *P. distasonis*, SPF mice were fed with a normal diet (LabDiet, USA, Cat# 5001, the same below) and treated with *P. distasonis* NSP007 ($2 \times 10^8$ CFU /mice/d) by gavage for seven days. Feces were collected on days 1 and 7 to measure the loads of *P. distasonis* ($n = 8$ mice in each group).

For the effect of *Parabacteroides distasonis* NSP007 on the production of NA, mice were fed with a normal diet and treated with $2 \times 10^8$ CFU/mice/d by gavage for seven days. Feces were collected on day -1 and day 7 to measure the NA level ($n = 12$ mice in each group).

For the effect of different bacteria strains (*Parabacteroides distasonis* NSP007, *Phocaeicola vulgatus*, *Bacteroides fragilis*, *Bacteroides thetaiotaomicron*, *Klebsiella pneumoniae*, *Escherichia coli*, and *Clostridium difficile*) on the production of NA, mice were fed with a normal diet and treated with $2 \times 10^8$ CFU/mice/d by gavage for seven days. Feces were collected on day -1 and day 7 to measure the NA level ($n = 6$ mice in each group).

## Metabolic assay
Three days before the mice were anesthetized with ether and killed by cervical dislocation, OGTT and ITT were performed on each group of animals. For the OGTT, glucose (1.5 g/kg body weight for mice) was administered orally after 6 h fasting, and blood samples were obtained from the tail tip at 0, 15, 30, 60, and 90 min. For the ITT, insulin (0.8 U/kg body weight for mice) was administered via an intraperitoneal injection after 6 h fasting, and blood samples were obtained from the tail tip at 0, 15, 30, 60, and 90 min. The glucose levels of tail vein blood samples were measured using a glucose analyzer (ACCU-CHEK Perform, Roche, Mannheim, Germany).

## Histopathological and immunohistochemical analysis
After euthanasia, the colonic tissues of mice were fixed in 4% paraformaldehyde. The liver tissues of the mice were fixed in 4% paraformaldehyde or embedded in an OCT compound. Paraformaldehyde-fixed, paraffin-embedded liver tissue slides were stained with H&E. The frozen liver sections were stained with Oil Red O. All processes were performed according to standard protocols followed by microscopy examination. The entire cross section was scanned and imaged in a digital pathology system (Leica Aperio LV1). For immunohistochemistry (IHC), paraffin sections were incubated with antibodies specific to Claudin1, Muc2, Occludin, ZO-1, and F4/80 [Anti-Claudin-1 (Abcam, cat. # ab307692, 1:100 for IHC), Anti-Muc2 (Abcam, cat. # ab272692, 1:2000 for IHC), Anti-Occludin (Cell Signaling Technology, cat. # 91131, 1:400 for IHC), Anti-ZO1 (Abcam, cat. # ab221547, 1:500 for IHC), Anti-F4/80 (Cell Signaling Technology, cat. # 70076S, 1:500 for IHC)]. The percentage of positive area was calculated using ImageJ 1.53k software.

## Biochemical analysis
The TNF-α and IL-1β levels were determined by an enzyme-linked immunosorbent assay (ELISA) kit (Fcmacs Biotech Co. Ltd., Nanjing, China) according to the manufacturer's instructions. Total cholesterol (TC), triglycerides (TG), free fatty acids (FFA), high-density lipoprotein cholesterol (HDL-C), low-density lipoprotein cholesterol (LDL-C), lipopolysaccharide (LPS), and insulin levels were measured by commercially available kit (Jiancheng Co. Ltd., Nanjing, China, Cusabio Co. Ltd., Wuhan, China, and Crystal Chem Inc., Illinois, USA) according to the manufacturer's instructions.

## 16S rRNA gene sequencing and data analysis
Total DNA was extracted from stool samples or fecal culture samples using a stool DNA extraction kit (TIANGEN Biotech Co.,Ltd, Beijing, China). The V4 region of the 16S rRNA gene was amplified by PCR from extracted and purified genomic DNA using 515 pairs of forward primers and 806 pairs of reverse primers. PCR amplification was performed on a PCR System (Bio-Rad, Hercules, CA, USA). The PCR amplification products were extracted separately from a 2% agarose gel and purified using an agarose gel DNA purification kit (TIANGEN Biotech Co.,Ltd, Beijing, China). The purified amplicons were quantified using a Qubit Fluorometer (Thermo Fisher Scientific, Waltham, MA, USA), pooled in equimolar quantities, and sequenced on an Illumina Miseq platform (Illumina, San Diego, CA, USA).

Raw sequencing data were processed with QIIME2 (version 2020.11). Paired-end reads were merged, trimmed, filtered, aligned, and clustered by amplicon sequence variants (ASV) using DADA2. Taxonomy was assigned using the Greengenes reference (v13.5) database. Analyses of alpha diversity, beta diversity, and bacterial taxonomic distribution were performed using MicrobiomeAnalyst (version 5.0).

## Gene expression analysis
Reverse transcription of total RNA was performed using the PrimeScript RT reagent kit (Takara Biomedical Technology Co., Ltd., Beijing, China). Quantitative real-time PCR was performed using TB Green Premix Ex Taq II (Takara Biomedical Technology Co., Ltd., Beijing, China) on a QuantStudio 7 Flex Real-Time PCR system (Thermo Fisher Scientific, Waltham, MA, USA). The primers were listed in the Supplementary Table 4. β-actin was used as an endogenous control. The relative mRNA expression levels were calculated using the $2^{-\triangle\triangle Ct}$ quantification method. All primers are provided by Sangon Biotech Co., Ltd. (Shanghai, China).

## Non-targeted metabolomics
The metabolites of bacteria culture supernatant were extracted according to the previously described method[55,56]. Bacterial culture supernatant (200 μL) was mixed with 200 μL ice-cold methanol and vortexed for 1 min. All samples were then sonicated in an ice water bath for 20 min and centrifuged at $16,000 \times g$, 4 °C for 20 min. The supernatants were collected and analyzed using a Shimadzu Nexera X2 UPLC system (Kyoto, Japan) coupled to an AB SCIEX Triple TOF 5600 mass spectrometer (AB SCIEX, Framingham, MA, USA)[57]. Chromatographic separation was performed at 40 °C with an ACQUITY UPLC HSS T3 column (2.1 mm × 100 mm, 1.8 μm). Acquired row mass data were processed using Progenesis QI software (version 2.0, Waters). The metabolic annotation was made by searching MS and MS/MS information against the HMDB database (version 4.0) and METLIN (version 1.0.6499.51447).

## Targeted quantification of NA and NAM
Feces, colonic contents, or bacteria culture supernatant were mixed with an extraction solvent (mentioned above) containing 1 μM chlorpropamide (internal standard). Feces or colonic contents (50 mg) were mixed with 400 μL ice-cold 80% methanol (v/v) and homogenized using KZ-II homogenizer (Servicebio, Wuhan, China). The extraction methods for bacteria culture supernatant were consistent with those described in "non-targeted metabolomics". Analysis of NA and NAM was performed by the liquid chromatography-tandem mass spectrometry (LC-MS/MS) system composed of a Shimadzu Nexera X2 high-performance liquid chromatography system (Kyoto, Japan) coupled to a Sciex 4500 triple quadrupole linear ion trap mass spectrometer (AB SCIEX, Framingham, MA, USA). Chromatographic separation was employed on an ACQUITY UPLC HSS T3 column (2.1 mm × 100 mm, 1.8 μm), and the mobile phase A contained 0.1% FA in water and mobile

phase B 0.1% FA in acetonitrile. All analytes were detected in multiple reaction monitoring (MRM) mode. Chromatographic separation was performed using a linear gradient as follows: 0–4 min, 5–40% B; 4–6 min, 40–80% B; 6–8 min, 80% B; 8–10 min, 80–5% B. Operational control of the LC-MS/MS was performed with Analyst version 1.6.2, and quantitative analysis was performed using MultiQuant software (version 3.0.1).

## Microbial strains

*P. distasonis* NSP007 (GuangDong Microbial Culture Collection Center, number: GDMCC61888), *Phocaeicola vulgatus*, *Bacteroides fragilis*, *Bacteroides thetaiotaomicron*, *Klebsiella pneumoniae*, *Escherichia coli*, and *Clostridium difficile* used in this study were isolated from the feces of patients with T2DM. Strains were cultured in an anaerobic chamber (Coy, Grass Lake, USA) with an atmosphere of 5% hydrogen, 5% carbon dioxide, and 90% nitrogen at 37 °C. It was grown in Brain Heart Infusion (BHI) with L-cysteine HCl (0.5 g/L), vitamin K1 (10 μl/L), hemin (5 mg/L), and resazurin (1 mg/L). The purity of the cultures was monitored by plating serial dilutions. All media, buffer, glass, and plasticware used in the study were exposed to anaerobic conditions at least 12 h before use.

## In vitro growth assays

For in vitro fermentation assays, human feces were collected and cultured as described[58]. Approximately 1 g of feces was resuspended in 10 mL of sterile anaerobic PBS, and the impurity was removed by centrifugation at $500 \times g$, 4 °C for 1 min. The bacteria pellets were collected by centrifugation at $8000 \times g$, 4 °C for 10 min resuspended in 10 mL of sterile anaerobic PBS, and incubated (1:9, v/v) in the adapted medium[59] under anaerobic conditions for 16 h (37 °C, 140 rpm) to activate the bacteria. The fecal inoculum was subsequently incubated (1:9, v/v) in the medium [containing 5 g/L carbon source (DOP, inulin, or β-glucan) or not] under anaerobic conditions for 48 h (37 °C; 140 rpm).

For *Parabacteroides* isolation and identification, the fermented samples were serially diluted in sterile PBS and spread on BHI agar plates [containing vancomycin (7.5 mg/L) and kanamycin (100 mg/L)]. After three days of incubation under anaerobic conditions at 37 °C, the colonies were picked, sub-cultured, and amplified by PCR using 27 forward and 1492 reverse primer pairs. The bacteria were subsequently identified using 16S ribosomal RNA sequencing, followed by BLAST analysis using the 16S rRNA database. The identification *P. distasonis* was cultured in BHI medium [DOP (5 g/L) as the sole carbon] for 24 h, and the $OD_{600}$ was measured by a Spectra Max 190 microplate reader (Molecular Devices Inc., San Jose, USA).

The total carbohydrate content was determined according to the previously described[60], with a slight modification. A total of 10 μL cell-free supernatants obtained through centrifugation at $8000 \times g$ for 15 min at 4 °C were added to 500 μL of ultrapure water, 500 μL of phenol, and 2 mL of sulfuric acid and then reacted at room temperature for 30 min. The absorbance value of the reaction mixture was determined at 490 nm using a Spectra Max 190 microplate reader.

To determine NA and NAM production by *P. distasonis* in vitro, *P. distasonis* were cultured overnight in BHI medium [supplemented with L-cysteine HCl (0.5 g/L), vitamin K1 (10 μl/L), hemin (5 mg/L) and resazurin (1 mg/L)] at 37 °C under anaerobic conditions, washed twice in PBS (containing 0.5 g/L L-cysteine HCl), and resuspended in PBS at a 1:1 dilution at 37 °C for 24 h. Samples were collected at the indicated time points. The NA and NAM levels were analyzed by LC-MS/MS.

To determine NA production by different bacteria strains (*Parabacteroides distasonis*, *Phocaeicola vulgatus*, *Bacteroides fragilis*, *Bacteroides thetaiotaomicron*, *Klebsiella pneumoniae*, *Escherichia coli*, and *Clostridium difficile*), strains were cultured overnight in BHI medium [supplemented with L-cysteine HCl (0.5 g/L), vitamin K1 (10 μl/L), hemin (5 mg/L) and resazurin (1 mg/L)] at 37 °C under anaerobic conditions, washed twice in PBS (containing 0.5 g/L L-cysteine HCl), and resuspended in PBS at a 1:1 dilution at 37 °C for 24 h. Samples were collected at the indicated time points. The NA levels were analyzed by LC-MS/MS.

To assess the effects of metformin, gliclazide, and glibenclamide (the top 3 administration antidiabetic drugs by the subjects in this study) on the growth of *P. distasonis*, after overnight activation, *P. distasonis* NSP007 were inoculated into BHI medium and incubated anaerobically at 37 °C for 24 h. The dosage of the drug is based on previous reports[61]. The $OD_{600}$ was measured by a Spectra Max 190 microplate reader.

## qPCR quantification of bacteria

Fecal DNA was extracted using the Tiangen stool DNA extraction kit according to the manufacturer's instructions. The primers listed in the Supplemental Table 4 were used for PCR-based amplification. Real-time PCR was performed using TB Green Premix Ex Taq II (Takara) on a QuantStudio 7 Flex Real-Time PCR system (Thermo Fisher Scientific). The PCR conditions included 50 °C for 2 min, then 95 °C for 30 s, followed by 40 cycles of denaturation at 95 °C for 20 s, annealing at 56 °C for 30 s, and extension at 72 °C for 30 s. Standard curves were constructed with reference bacteria as previously described[62]. All primers are provided by Sangon Biotech Co., Ltd. (Shanghai, China).

## Cell culture

Human Caco-2 cells were purchased from the American Type Culture Collection (ATCC). Cells were maintained in Dulbecco's modified Eagle's medium (DMEM) supplemented with 10% fetal bovine serum (FBS), and antibiotics (100 U/mL penicillin and 100 μg/mL streptomycin sulfate) at 37 °C with 5% $CO_2$ in a humidified incubator before treatment.

## Caco-2 cells treatment by NA and MPN

For the NA efficiency assay in Caco-2 cells, cells were grown to 80 to 90% confluence and then incubated for 12 h with serum-free medium (Control) or NA (5, 10, 50, and 100 μM) in the upper layer. For the NA and MPN efficiency assay in Caco-2 cells, cells were grown until 80 to 90%, then pretreated with serum-free medium (Control) or MPN (1 μM) in the upper layer for 1 h and subsequently stimulated with NA (100 μM) for an additional 12 h. Cells were collected after lysis by Trizol lysate for qPCR.

## Statistical analysis

Parametric or non-parametric statistical tests were appropriately applied after testing for the normal data distribution. For comparison between the two groups, an unpaired *t*-test (with equal variances) or an unpaired *t*-test with Welch's correction (with different variances) was used when samples were normally distributed, and the Mann–Whitney U test was used when samples were not normally distributed. For comparisons among more than two groups, one-way ANOVA, followed by Tukey's (with equal variances) or Dunnett's T3 (with different variances) post hoc test was used when samples were normally distributed, and Kruskal–Wallis was used when samples were not normally distributed. Statistical analyses were performed using SPSS (version 26.0). No data were excluded during the data analysis. All data is representative of two or more independent experiments and got similar results.

## Reporting summary

Further information on research design is available in the Nature Portfolio Reporting Summary linked to this article.

## Data availability

The 16S rRNA data generated in this study were deposited in the NCBI SRA database (https://www.ncbi.nlm.nih.gov/bioproject/1030208, accession code PRJNA1030208). The mass spectrometry data generated in this study were deposited in the MetaboLights (accession code MTBLS7943). All other data generated or analyzed during this study are included in this published article (and its supplementary information files). Source data are provided with this paper.

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

## Acknowledgements

This work was supported by the National Natural Science Foundation of China for Distinguished Young Scholars (31825020, S.P.N.), the Technological Project of Jiangxi Province (20232BCD44003, S.P.N.), the Technological Innovation Guidance Science and Technology Project of Jiangxi Province (20203AEI007, S.P.N.), the Key Technological Project of Jiangxi Province (20212AAF01005, S.P.N.), and Key Laboratory of Bioactive Polysaccharides of Jiangxi Province (20212BCD42016, S.P.N.).

## Author contributions

Conceptualization, Y.G.S., Q.X.N., and S.P.N.; methodology, Y.G.S., Q.X.N., S.S.Z., H.J.H., S.Z., C.H.C., J.R.Y.; software, Y.G.S., Q.X.N., S.S.Z., investigation, H.H.C., S.S.Z., J.L.H., S.L., J.B.C., B.J.Z., Z.T.Z., S.J.P.; resources, H.H.C., J.L.H., P.H., L.L., S.P.N; writing – original draft, Y.G.S., Q.X.N.; writing – review & editing, S.P.N.; visualization, Y.G.S., Q.X.N., S.S.Z.; supervision, S.P.N.; project administration, S.P.N.; funding acquisition, S.P.N.

## Competing interests

The authors declare no competing interests.
