## [Peer Review File · Nature Communications]

REVIEWER COMMENTS

Reviewer #1 (Remarks to the Author):

In this study, authors identified *Parabacteroides distasonis* as inversely correlated to insulin resistance in type 2 diabetes patients in two independent Chinese cohorts. The amelioration of obesity and insulin resistance was mediated by nicotinic acid-GPR109a axis. The study also demonstrated that DOP intervention resulted in the increase of *P. distasonis* and nicotinic acid to control obesity and insulin resistance. This study is potentially interesting; however, the reviewer has some concerns as follows.

Major points

1. It is known that many bacteria have potential to produce nicotinic acid (Front. Genet. 6:148, 2015), and human data in this study showed that *P. distasonis* does not account for a significant proportion. It would be necessary to confirm the involvement of nicotinic acid production by other bacteria.
2. Related to #1, is the effect not observed when other bacteria are administered?
3. Do the effects of *P. distasonis* only depend on GPR109a? Are the effects cancelled by treatment of GPR109a inhibitor or in GPR109a knockout mice?
4. Experiments in Fig. 5J show that administration of *P. distasonis* increases nicotinic acid, but in some mice showed the reduction. No effects were predicted in these mice, but the data in Fig. 4 show a beneficial effect in all mice. Similarly, although there are statistical differences in nicotinic acid in between HC and T2DM (Fig. 5N), most individuals have similar values in HC and T2DM. This is assumed to be due to effective molecules other than nicotinic acid. Identification of this molecule and elucidation of its mechanism would be an excellent.
5. Administration of *P. distasonis*, DOP, and nicotinic acid into mice suppressed HFD-induced obesity and insulin resistance. As the underlining mechanism, authors focused on intestinal improvement of barrier function via activation of GPR109a. It is reasonable that insulin resistance can be improved by intestinal barrier function, but are there similar mechanisms for obesity and fat accumulation?
6. How nicotinic acid increased the GPR109a expression?
7. Administration of DOP increased relative abundance of *P. distasonis* in in vitro fecal culture and mice. How does *P. distasonis* abundance increase? For examples, does *P. distasonis* utilize directly DOP or DOP-derived metabolites from other intestinal bacteria for the growth of *P. distasonis*? Relatedly, does DOP influence metabolic functions of *P. distasonis* such as producing nicotinic acid?
8. In human, do dietary intakes of fiber and/or nicotinic acid influence relative abundance of *P. distasonis* and its function such as producing nicotinic acid? Also, is the dependence on nicotinic acid

from *P. distasonis* different between those who get a sufficient amount of nicotinic acid from their diet and those who do not get enough?

Minor points

1. In Fig. 7C, is “ $r = -0.3780$ ” correct? It looks positive correlation.
2. In lines 477-481, please describe the name of kits and/or companies.
3. In 575-583, please describe primer sequences for detecting *P. distasonis*.

Reviewer #2 (Remarks to the Author):

In their study, Sun and colleagues elegantly decipher the mechanisms by which dietary fiber can improve insulin resistance through microbiota modulation. The manuscript compiles a large body of work, including studies with two cohorts of humans and numerous mechanistic studies with mice and in vitro studies. In that sense, I congratulate the authors for their excellent work. In addition, as a result, the work provides a new mechanism involving the nicotinic acid metabolite and the GPR109a receptor.

However, there are some issues to address that are necessary to ensure its successful publication in Nat Com.

Reviewer #3 (Remarks to the Author):

The manuscript by Dr. Yonggan Sun and colleagues comprises preclinical and clinical analyses.

From preclinical studies, investigators report that a gut bacterial strain, *Parabacteroides distasonis* NSP007, when orally delivered to mice over a short period (a week) improves measures of insulin sensitivity and diminish markers of systemic inflammation, in part via a strengthened intestinal integrity.

Mechanistically, the authors provide evidence that *P. distasonis*-derived nicotinic acid (NA) enhances intestinal barrier function through an activation of the intestinal G-protein-Coupled Receptor 109a (GPR109a), a mechanism that putatively may contribute to an enhancement of insulin sensitivity.

In a short-term intervention in mice, the investigators show that feeding a specific dietary fiber, dendrobium officinale polysaccharide (DOP), stimulates growth of *P. distasonis* along with an improvement of insulin sensitivity.

From the outcome of clinical analyses (cross-sectional data of two small human cohorts), it is concluded that abundance of *P. distasonis* associates inversely with a surrogate measure of insulin resistance.

Major general comments and recommendations.

Studies of the potential role of *P. distasonis* strains for human health and diseases is currently ongoing in many laboratories, and no clear picture can be drawn (GUT MICROBES 2021, VOL. 13, NO. 1, e1922241 doi.org/10.1080/19490976.2021.1922241).

Thus, *P. distasonis* has been shown to have both beneficial and detrimental effects on metabolic dysfunctions including diabetes, complicating its definition as a beneficial commensal or a pathogenic gut bacterium (Cell Rep. 2019 Jan;26(1):222–235.e5.

doi:10.1016/j.celrep.2018.12.028;Front Cell Infect Microbiol. 2020;10:188.

doi:10.3389/fcimb.2020.00188).

On this background there is a major need to clarify whether the host effects of bioactive molecules that are synthesized by this bacterial species is strain- and context-dependent.

The present preclinical studies add to this clarification showing suggestive experimental evidence that in relation to glucose metabolism in mice, *P. distasonis* strain NSP007 via release of NA may be host beneficial and that the abundance of the strain is contextual. In this case, dependent on the amount of specific dietary fiber.

In contrast, results from the present clinical studies are far from being compelling and convincing due to the very small number of individuals included (60 x 2). This reviewer suggests that cohorts of many hundreds of well-characterized non-diabetic individuals are included with measures of qPCR – quantified stool abundance of *P. distasonis* strain NSP007 as well as *P. distasonis* species, measures of whole body insulin sensitivity, a panel of systemic inflammation plasma markers and plasma markers of intestinal integrity. Such an extended clinical study protocol will allow for explorations of correlations between a specific *P. distasonis* strain versus the multiple strains of the same *P. distasonis* species and measures of insulin sensitivity, gut permeability and inflammation.

Specifics

- 1) Throughout the manuscript, it should be made clear whether the analyses deal with *P. distasonis* strain NSP007 or *P. distasonis* species. Lines 800-801 of the manuscript indicate that the *P. distasonis* species may comprise 43 strains. Do all 43 strains produce NA?
- 2) All data from diabetics should be adjusted for intake of drugs.
- 3) In oral gavage mice studies, please provide level of engraftment of *P. distasonis*.
- 4) Give specificity, sensitivity and intra-coefficient of variation for the NA assay.

REVIEWER COMMENTS

Reviewer #1 (Remarks to the Author):

In this study, authors identified *Parabacteroides distasonis* as inversely correlated to insulin resistance in type 2 diabetes patients in two independent Chinese cohorts. The amelioration of obesity and insulin resistance was mediated by nicotinic acid-GPR109a axis. The study also demonstrated that DOP intervention resulted in the increase of *P. distasonis* and nicotinic acid to control obesity and insulin resistance. This study is potentially interesting; however, the reviewer has some concerns as follows.

Response: We appreciate the positive comments and a series of helpful suggestions, and addressed all comments and concerns raised by the reviewers. Please see below for our point-by-point response.

Major points

1. It is known that many bacteria have potential to produce nicotinic acid (Front. Genet. 6:148, 2015), and human data in this study showed that *P. distasonis* does not account for a significant proportion. It would be necessary to confirm the involvement of nicotinic acid production by other bacteria.

Response: Thanks for your reminder, stefanía et al.'s (Front. Genet. 6:148, 2015) report focuses primarily on the converging pathways of niacin and lists some bacteria that may produce niacin acid, but *P. distasonis* is not included in their research. Therefore, we conducted experiments to assess the ability of both *P. distasonis* and those strains they identified as having niacin acid-producing capacity to produce niacin acid *in vitro* and *in vivo*. The results showed that all tested strains were able to produce nicotinic acid *in vitro*, whereas *P. distasonis* possessed the best nicotinic acid-producing ability *in vivo* (Response figure 1). Furthermore, a recent study has also found that *P. distasonis* significantly increases the levels of nicotinic acid in BHI culture (Nat Commun. 2023;14(1):1829). Taken together, we considered that *P. distasonis*-derived nicotinic acid is important to improve insulin resistance.

Response figure. 1 Assessment of nicotinic acid-producing ability of different bacteria strains both *in vitro* and *in vivo*. (a) The intensity of nicotinic acid in the culture supernatant of bacteria strains after 24 h of incubation, n = 3 per group. (b-d) Level of fecal nicotinic acid after different bacteria strains treatment. Mice were treated with different strains (2×10^8 CFU /mice/d) for 7 days. n = 6 mice per group. (b) Experimental schema of mouse experiments, (c) Relative abundance of different strains before and after 7 days oral gavage, (d) Change of fecal nicotinic acid after different bacteria strains treatment.

This information has been added to lines 262-266 and Supplementary Fig. 7k-n in the revised manuscript.

2. Related to #1, is the effect not observed when other bacteria are administered?

Response: We performed an animal experiment to investigate the change of nicotinic acid by different bacteria strains treatment (**Response figure. 1b**). The results demonstrate that *P. distasonis* exhibits a favorable colonization ability among these strains (**Response figure. 1c**). Additionally, the level of fecal nicotinic acid following *P. distasonis* treatment was found to be higher compared to other strains (**Response figure. 1d**), indicating that *P. distasonis* may play an important contribution to the nicotinic acid level in the host.

3. Do the effects of *P. distasonis* only depend on GPR109a? Are the effects cancelled by treatment of GPR109a inhibitor or in GPR109a knockout mice?

Response: Thanks for your comment. To investigate whether the beneficial effects of *P. distasonis* in improving IR are dependent on the activation of GPR109a, mice were pretreated with HFD for eight weeks, with *P. distasonis*, MPN (an inhibitor of GPR109a), and *P. distasonis* +MPN for a further five weeks (**Response figure. 2a**). The supplementation with MPN eliminated the improvement of *P. distasonis* on weight gain, fat accumulation, IR, chronic inflammation, and intestinal permeability in IR mice (**Response figure. 2b-t**).

Response figure. 2 *P. distasonis* ameliorates IR in a GPR109a activation-dependent manner

(a) Experimental scheme for (b to t). Mice were fed an HFD for 8 weeks and then treated with PBS, *P. distasonis*, MPN, and *P. distasonis*+MPN for a further five weeks.

(b) *P. distasonis* abundance in feces accessed by qPCR.

(c) Body weight curve.

(d) Body weight change (%).

(e) Food intake.

(f) Liver weight.

(g) Epididymal fat weight.

(h) Epididymal fat/body weight (%).

(i-j) OGTT (i) and AUC (j).

(k-l) ITT (k) and AUC (l).

(m) Fasting insulin level.

- (n) HOMA-IR.
- (o) Serum TNF- α .
- (p) Serum IL-1 β .
- (q) Serum LPS.
- (r) Relative mRNA expression of genes related to intestinal permeability in colon.
- (s) Relative mRNA expression of *Gpr109a* in colon.
- (t) Relative mRNA expression of genes related to inflammation in liver.

Data are presented as the mean \pm SEM. n = 6–8 mice per group. Statistical analysis was performed using One-way ANOVA with Tukey's post hoc test for (b), (f), (g), (h), (i), (k), (l), (m), (n), (o), (p), (q), *Zo1* of (r), (s), and (t), One-way ANOVA with Dunnett's T3 post hoc test for (c), (j), *Claudin1*, and *Occludin* of (r), Kruskal-Wallis test for (d), and *Muc2* of (r). *, $P < 0.05$; **, $P < 0.01$.

This information has been added to lines 302-308 and Supplementary Fig. 9 in the revised manuscript.

4. Experiments in Fig. 5J show that administration of *P. distasonis* increases nicotinic acid, but in some mice showed the reduction. No effects were predicted in these mice, but the data in Fig. 4 show a beneficial effect in all mice. Similarly, although there are statistical differences in nicotinic acid in between HC and T2DM (Fig. 5N), most individuals have similar values in HC and T2DM. This is assumed to be due to effective molecules other than nicotinic acid. Identification of this molecule and elucidation of its mechanism would be an excellent.

Response: Thank you for your suggestion. In **Fig. 5j**, the level of fecal nicotinic acid showed a slight decrease in 2 normal SPF mice. We speculate that this may be due to unsuccessful colonization of *P. distasonis*, and we found the abundance of fecal *P. distasonis* was not changed in these 2 mice (**Response figure. 3c**). Furthermore, we replicated this experiment with an additional 12 mice, and found only one mouse was a failure successfully colonization *P. distasonis* and production of nicotinic acid, this difference may be caused by the heterogeneity of the host (**Response figure. 3d, e**).

Regarding the results in humans, as you mentioned, not all the HC had higher nicotinic acid levels compared to T2DM. This could be a common phenomenon observed in human cohort studies, where not all control group fecal metabolites are higher or lower than the model group (**Nat Med. 2018;24(12):1919-1929**, **Nat Med. 2019;25(8):1225-1233**). The levels of metabolites in human feces may influenced by various factors such as diet, genes, physiological status, and gut microbiota composition (**Cell Host Microbe. 2021;29(3):394-407**, **Annu Rev Pathol. 2020;15:345-369**). We found that nicotinic acid may be an important factor in improving IR through clinical variation, and illustrated the importance of *P. distasonis*-nicotinic acid-GPR109a pathway on insulin resistance, which also provides an explanation for some of the previous clinical studies (**Diabetes Care. 2018;41(3):398-405**). In the clinical study, nicotinic acid can ameliorate IR in humans, but the detailed mechanism is not clear.

Response figure. 3 Colonization of *P. distasonis* NSP007 and production of nicotinic acid in mice.

Previous results: (a) Experimental schema of mouse experiments, (b) Fecal nicotinic acid level, n = 8 mice per group, (c) Fecal *P. distasonis* abundance analyses of mice by qPCR, n = 8 mice per group.

New results: (d) Experimental schema of mouse experiments, (e) Fecal *P. distasonis* abundance analyses of mice by qPCR, n = 12 mice per group, (f) Fecal nicotinic acid level, n = 12 mice per group.

Response figure 3d-f has been supplemented in the revised manuscript (Fig. 5i-k).

5. Administration of *P. distasonis*, DOP, and nicotinic acid into mice suppressed HFD-induced obesity and insulin resistance. As the underlining mechanism, authors focused on intestinal improvement of barrier function via activation of GPR109a. It is reasonable that insulin resistance can be improved by intestinal barrier function, but are there similar mechanisms for obesity and fat accumulation?

Response: A long-term high-fat diet impairs intestinal barrier function, leading to chronic inflammation caused by the translocation of bacterial LPS into the bloodstream (Trends Endocrinol Metab. 2022;33(4):247-265, PLoS One. 2012;7(10):e47713). Long-term chronic inflammation is a significant factor in obesity, fat accumulation, and insulin resistance (Physiology (Bethesda). 2016;31(4):283-93, Cell. 2013;152(4):673-84). Therefore, improving intestinal barrier function could reduce the chronic inflammation caused by LPS leakage and its associated fat accumulation and obesity (Nat Med. 2019;25(7):1096-1103, Gut. 2019;68(2):248-262). Our research also demonstrates that *P. distasonis* NSP007, DOP, and nicotinic acid not only improve intestinal barrier function and insulin resistance but also have beneficial effects on mouse weight and fat accumulation (Fig. 3c; Supplementary Fig. 3c; Fig. 4d; Supplementary Fig. 6c; Fig. 6i; Supplementary Fig. 8c).

6. How nicotinic acid increased the GPR109a expression?

Response: High-fat diet results in the decreased expression of GPR109a in the intestines, and nicotinic acid can enhance the expression of GPR109a in obese mice (FASEB J. 2019;33(4):4765-4779). Numerous studies have reported that nicotinic acid is a potent agonist of GPR109a (Nat Med. 2003;9(3):352-5, J Biol Chem. 2005;280(29):26649-52, J Clin Invest. 2005;115(12):3634-40). Administration of extracts from the flower buds of honeysuckle, pagoda tree, and gardenia can enhance the expression of GPR109a in the mouse gut and improve intestinal barrier function (Pharmacol Res. 2020;159:104809). Additionally, the inclusion of legumes in the diet can increase the expression of GPR109a in the mouse gut and improve intestinal barrier function (J Nutr Biochem. 2017;49:89-100). These studies collectively suggest that activation of GPR109a could improve intestinal barrier function. However, direct evidence of NA promoting the expression of GPR109a is currently lacking, we speculate that it may be ligand-induced receptor expression, but the detailed mechanism is unclear at present.

7. Administration of DOP increased relative abundance of *P. distasonis* in in vitro fecal culture and mice. How does *P. distasonis* abundance increase? For examples, does *P. distasonis* utilize directly DOP or DOP-derived metabolites from other intestinal bacteria for the growth of *P. distasonis*? Relatedly, does DOP influence metabolic functions of *P. distasonis* such as producing nicotinic acid?

Response: To the best of our knowledge, there are currently no studies reports that *P. distasonis* can degrade DOP. Our research indicated that DOP has no significant effects on the proliferation of *P. distasonis* in vitro (Response figure. 4a, b). Recent research has revealed that a Ganoderma meroterpene derivative (GMD) improves atherosclerosis by promoting the proliferation of *P. merdae* (belongs to the *Parabacteroides* as *P. distasonis*), but the growth of *P. merdae* was not influenced by the addition of GMD in YCFA medium in vitro. They deduced that the enrichment of *P. merdae* in the gut by GMD resulted from its indirect impact on the gut microbiome (Nat Metab. 2022;4(10):1271-1286). Therefore, we considered the growth of *P. distasonis* may depend on the metabolites produced by other gut microbiota from the degradation of DOP. Additionally, we further investigate the impact of DOP on the ability of *P. distasonis* to produce nicotinic acid, the results showing that DOP has no influence on nicotinic acid-producing ability of *P. distasonis* (Response figure. 4c, d). Therefore, we believe that the increase of nicotinic acid by DOP is primarily achieved by upregulating the abundance of *P. distasonis*.

Response figure. 4 Effect of DOP on the growth of *P. distasonis* NSP007 and its nicotinic acid-producing ability (a) Experimental schema, (b) Growth curve of *P. distasonis* NSP007 with DOP or not (n = 3 per group). (c-d) The intensity of nicotinic acid in the (c) PBS or (d) YCFA medium supernatant of *P. distasonis* NSP007 at 24 h (n = 3 per group).

8. In human, do dietary intakes of fiber and/or nicotinic acid influence relative abundance of *P. distasonis* and its function such as producing nicotinic acid? Also, is the dependence on nicotinic acid from *P. distasonis* different between those who get a sufficient amount of nicotinic acid from their diet and those who do not get enough?

Response: Thanks for your comment. It has been shown that dietary fiber supplementation could increase the abundance of *Parabacteroides/P. distasonis* in humans (**Nat Med. 2021;27(7):1272-1279, Cell Host Microbe. 2020;27(3):389-404.e6**), but there are no reports on the nicotinic acid intake affect the *P. distasonis* abundance and its ability to produce nicotinic acid in humans. Our research group is conducting relevant clinical trials, which may solve the problem (such as the role of DOP on *P. distasonis* abundance) in the future. Furthermore, supplementation with nicotinic acid could improve IR in humans (**Diabetes Care. 2018;41(3):398-405**). Based on the literature and available data, we believe that dietary or microbiota-derived nicotinic acid both has a beneficial effect on IR. If dietary nicotinic acid intake is relatively insufficient, nicotinic acid produced by the gut microbiota may provide health benefits to the host.

Minor points

1. In Fig. 7C, is “ $r = -0.3780$ ” correct? It looks positive correlation.

Response: Thanks for your reminder, we have corrected these results. (**Fig. 7e**)

2. In lines 477-481, please describe the name of kits and/or companies.

Response: Thanks for your reminder, we have supplemented this description. (**lines 504-509**)

3. In 575-583, please describe primer sequences for detecting *P. distasonis*.

Response: Thanks for your reminder, we have listed all primers used in **Supplemental Table 2**.

Reviewer #2 (Remarks to the Author):

In their study, Sun and colleagues elegantly decipher the mechanisms by which dietary fiber can improve insulin resistance through microbiota modulation. The manuscript compiles a large body of work, including studies with two cohorts of humans and numerous mechanistic studies with mice and in vitro studies. In that sense, I congratulate the authors for their excellent work. In addition, as a result, the work provides a new mechanism involving the nicotinic acid metabolite and the GPR109a receptor.

However, there are some issues to address that are necessary to ensure its successful publication in Nat Com.

Response: We appreciate the positive comments and a series of helpful suggestions, and addressed all comments and concerns raised by the reviewers. Please see below for our point-by-point response.

MAYOR COMMENTS

Line 92: Please, explain the reasoning to analyze only the top 25 bacterial genera

Response: We focus our research on the genus abundance of approximately 90% (top 25 bacterial genera) of the gut microbiota, as we think these genera could effectively represent the change of gut microbiota. Additionally, several studies also put more attention on the genus abundance of the top 15 (Nat Med. 2018;24(12):1919-1929) or 50 (Nature. 2018;562(7728):583-588) of gut microbiota. While we acknowledge the importance of the remaining 10% of bacterial genera, given the limitations of our study (16S rRNA gene amplicon sequencing), we focused our analysis on the 90%.

Fig 1H: There is an issue with the colors in Fig H. Looking at the figure, black dots belong to the control group.

Response: Thanks for your reminder, we have corrected these results. (Fig. 1h)

Fig 2G: Same problem with the colors that the one described above occurs in Fig2G.

Response: Thanks for your reminder, in the Fig2G, DOP reduced the abundance of *P. merdae*, *P. johnsonii*, *P. goldsteinii*, and *P. gordonii*. (Fig. 2g)

Fig2G-F: Why if it is a qPCR the data is expressed as relative abundance?

Response: Thanks for your reminder, we want to show the proportion of target bacteria in gut microbiota. Moreover, previous studies have used the method to show this data (Nature. 2022;610(7932):562-568; Nat Metab. 2022;4(10):1271-1286).

In the experiment where mice were treated with a HFD diet plus DOP, can the authors justify the absence of a control group (mice fed a control diet). This group might have been useful to evaluate the magnitude of the improvements caused by DOP

Response: Thanks for your reminder, we have now included the data as you mentioned. (Fig. 3a-n, and Supplementary Fig. 3a-i).

Line 218: Why did the authors use samples for an *in vitro* assay rather than the cecal content of mice treated with DOP for the first metabolomic analyses?

Response: Thanks for your comment, there are many metabolic interaction patterns among gut microbiota, and the host also affects the structure and metabolism of gut microbiota (Cell Host Microbe. 2023;31(4):485-499, Cell Metab. 2021;33(10):1988-2003.e7). Therefore, the culture of strains *in vitro* can eliminate the influence from host and the other gut microbiota, which is more conducive to discovering the metabolic characteristics of the strain to a certain extent. However, the results from *in vitro* and *in vivo* studies must be mutually combined and validated.

Line 449-450: Mice were killed after the GTT and IIT test. How did the authors manage the animal stress? Do they consider that stress might have influence the outcomes of the study (for instance, pro-inflammatory markers)?

Response: We are sorry that this was a misunderstanding caused by our misdescription, and we have corrected it in the manuscript. (Lines 481-482)

“Three days before the mice were anesthetized with ether and killed by cervical dislocation, OGTT and ITT were performed on each group of animals.”

MINOR COMMENTS

Line 42: Please, the specify the tissue or organ producer of TNF after the TLR4 activation.

Response: We have corrected this description. (Lines 39-40)

Line 96: Spell “HC”

Response: We have corrected this description. (Line 95)

Line 98: The p-value shown in the text and in the graph are not matching

Response: We have corrected this description. (Line 97)

Line 107 and line 112: Replace “strains” by “genera”

Response: Thanks for your reminder, through our careful confirmation, we consider this description has no mistakes.

Line 118: Spell “IN” as insulin. The same for “BG”

Response: We have corrected this description. (Lines 114, 118)

Line 130: Remove the word “fecal”

Response: We have corrected this description. (Line 130)

Line 169: The correlation between *P. distasonis* and TNF is missing in the text.

Response: We have corrected this description. (Line 168)

Line 175: Please, rephrase the sentence. It seems that two different animal models (antibiotic and GFM) have been used.

Response: We have corrected this description. (Line 174)

Line 196: To be consistent, please replace “PD” by “*P. distasonis*”.

Response: We have corrected this description. (Line 194)

Line 197: One introductory sentence, like in the previous sections, would help to the understand better the experiment.

Response: We have corrected this description. (Lines 195-197)

Line 202: Spell “IHTG”

Response: We have corrected this description. (Line 203)

Line 536: The human donor where *P. distasonis* was isolated is a heathy subject?

Response: *P. distasonis* NSP007 was isolated from the feces of patients with T2DM, and we have corrected this description. (Line 567)

Reviewer #3 (Remarks to the Author):

The manuscript by Dr. Yonggan Sun and colleagues comprises preclinical and clinical analyses.

From preclinical studies, investigators report that a gut bacterial strain, *Parabacteroides distasonis* NSP007, when orally delivered to mice over a short period (a week) improves measures of insulin sensitivity and diminish markers of systemic inflammation, in part via a strengthened intestinal integrity.

Mechanistically, the authors provide evidence that *P. distasonis*-derived nicotinic acid (NA) enhances intestinal barrier function through an activation of the intestinal G-protein-Coupled Receptor 109a (GPR109a), a mechanism that putatively may contribute to an enhancement of insulin sensitivity.

In a short-term intervention in mice, the investigators show that feeding a specific dietary fiber, dendrobium officinale polysaccharide (DOP), stimulates growth of *P. distasonis* along with an improvement of insulin sensitivity. From the outcome of clinical analyses (cross-sectional data of two small human cohorts), it is concluded that abundance of *P. distasonis* associates inversely with a surrogate measure of insulin resistance.

Response: We appreciate the insightful comments and a series of helpful suggestions, and addressed all comments and concerns raised by the reviewers.

Major general comments and recommendations.

Studies of the potential role of *P. distasonis* strains for human health and diseases is currently ongoing in many laboratories, and no clear picture can be drawn (GUT MICROBES 2021, VOL. 13, NO. 1, e1922241 doi.org/10.1080/19490976.2021.1922241). Thus, *P. distasonis* has been shown to have both beneficial and detrimental effects on metabolic dysfunctions including diabetes, complicating its definition as a beneficial commensal or a pathogenic gut bacterium (Cell Rep. 2019 Jan;26(1):222–235.e5. doi:10.1016/j.celrep.2018.12.028; Front Cell Infect Microbiol. 2020;10:188. doi:10.3389/fcimb.2020.00188). On this background there is a major need to clarify whether the host effects of bioactive molecules that are synthesized by this bacterial species is strain- and context-dependent.

Response: Thanks for your comment, and there are indeed some negative reports on the effects of *P. distasonis* on disease. However, the role of the gut microbiota may vary across different disease states (Nature. 2019;569(7758):655-662; Nature. 2019;569(7758):663-671) based on the reports related to integrative human microbiome project (iHMP) (Cell Host Microbe. 2014;16(3):276-89). To our knowledge, reports of *P. distasonis* in aggravating metabolic diseases are more limited. As you mentioned, Wang et al. (Cell Rep. 2019;26(1):222–235.e5) reported that *P. distasonis* (isolated from mice) could ameliorate metabolic disease (focus on the effect on obesity) via bile acid-FXR axis, especially the UDCA and LCA. Our data showed that *P. distasonis* NSP007 (isolated from humans) has no effect on the levels of LCA and UDCA in the ileal contents (Response Figure. 5a, b), nor the mRNA expression of FXR and FGF15 in the ileal tissues of mice (Response Figure. 5c, d). Therefore, we considered that the improvement of metabolic diseases by *P. distasonis* may be strain-specific.

Response Figure. 5 Effect of *P. distasonis* NSP007 on bile acid metabolism in IR mice.

(a-b) The contents of (a) LCA and (b) UDCA in the ileal contents of mice, (c-d) Relative expression of intestinal *Fxr* (c) and *Fgf15* (d) in the ileum of mice. Mice treated with *P. distasonis* NSP007 for 5 weeks. n = 6 mice per group.

The present preclinical studies add to this clarification showing suggestive experimental evidence that in relation to glucose metabolism in mice, *P. distasonis* strain NSP007 via release of NA may be most beneficial and that the abundance of the strain is contextual. In this case, dependent on the amount of specific dietary fiber. In contrast, results from the present clinical studies are far from being compelling and convincing due to the very small number of individuals included (60 x 2). This reviewer suggests that cohorts of many hundreds of well-characterized non-diabetic individuals are included with measures of qPCR –quantified stool abundance of *P. distasonis* strain NSP007 as well as *P. distasonis* species, measures of whole body insulin sensitivity, a panel of systemic inflammation plasma markers and plasma markers of intestinal integrity. Such an extended clinical study protocol will allow for explorations of correlations between a specific *P. distasonis* strain versus the multiple strains of the same *P. distasonis* species and measures of insulin sensitivity, gut permeability and inflammation.

Response: Thank you for your reminders. Indeed, the larger the sample size, the more reliable the data obtained. However, recruiting such a large cohort within a short time is very difficult for us. We are well aware of this issue and try to address it in our paper by analyzing the relationship between *Parabacteroides* and IR phenotype using the data from the Human Microbiome Project database (<https://www.hmpdacc.org/hmp/>) (Fig. 7i-1). We are also actively seeking additional relevant datasets to conduct further analysis, but unfortunately, we have not yet found a suitable dataset.

Specifics

1) Throughout the manuscript, it should be made clear whether the analyses deal with *P. distasonis* strain NSP007 or *P. distasonis* species.

Response: We have corrected this description throughout this revised manuscript.

2) Lines 800-801 of the manuscript indicate that the *P. distasonis* species may comprise 43 strains. Do all 43 strains produce NA?

Response: We evaluated the capacity of 44 *P. distasonis* (42 *P. distasonis*, 1 *P. distasonis* NSP007, and 1 purchased *P. distasonis* ATCC8503) on the production of NA. The data showed that all *P. distasonis* strains could produce NA (Response Figure. 6).

Response Figure. 6 The intensity of NA in the culture supernatant of 44 *P. distasonis* strains at 24 h.

This information has been added to lines 243-245 and Supplementary Fig. 7e in the revised manuscript.

3) All data from diabetics should be **adjusted** for intake of drugs.

Response: Thanks to your suggestion, we reviewed the data on drug administration (**Response Table. 1**), which showed that many T2DM patients have taken more than one anti-diabetic medication. Therefore, we think it's inapplicable to adjust data by drug information. However, to examine the effect of drugs on the data, we investigate the effect of the top 3 administration drugs (metformin, gliclazide, and glibenclamide) on the growth of *P. distasonis* NSP007. The results showed that the three drugs had no significant effect on the growth of *P. distasonis* (**Response Figure. 7**).

Response Table. 1 Drug use of subjects

Characteristics	Cohort 1 T2DM (n = 30)	Cohort 2 T2DM (n = 30)
Oral antidiabetic drugs, %	26 (86.7)	23 (76.7)
*Metformin, %	23 (76.7)	20 (66.7)
*Glibenclamide, %	7 (23.3)	8 (26.7)
*Gliclazide, %	7 (23.3)	3 (10.0)
*Other drugs, %	5 (16.7)	3 (10.0)

*Some patients take more than one antidiabetic drug.

This information has been added to Supplementary Table 1 and 3 in the revised manuscript.

Response Figure. 7 Effects of three main antidiabetic drugs taken by T2DM patients on the growth of *P. distasonis*. (a) Metformin treatment, (b) Glibenclamide treatment, and (c) Gliclazide treatment.

Data are presented as the mean ± SEM. n = 3 per group.

Method: After overnight activation, *P. distasonis* was inoculated into BHI medium and incubated anaerobically at 37 °C for 24 h. The dosage of the drug is based on previous reports (Nature. 2018;555(7698):623-628). The OD₆₀₀ was measured by a Spectra Max 190 microplate reader (Molecular Devices Inc.).

This information has been added to line 315-319 and Supplementary Fig. 10 in the revised manuscript.

4) In oral gavage mice studies, please provide level of engraftment of *P. distasonis*.

Response: Thanks for your comment, our data showed that *P. distasonis* NSP007 can successfully colonize in the mouse intestines, regardless of whether they were fed a high-fat diet (Fig. 4b) or a normal Chow diet (Fig. 5j).

5) Give specificity, sensitivity and intra-coefficient of variation for the NA assay.

Response: Based on a series of methodology studies for NA detection, we found the limits of detection (LOD) is 0.23 ng/mL, recovery of NA was in the range of 97.4%—103.1% at three spiked levels, with relative standard deviations (RSDs) of 2.5%—4.3%.

Response Table. 2 Result of spiked recoveries and relative standard deviations

Compound Name	Concentration (mg/g)	Spiked Concentration (mg/g)	Determined Concentration (mg/g)						Recovery (%)	RSD (%)
			1	2	3	4	5	6		
Nicotinic acid	1.00	1.00	4.05	4.09	4.11	4.16	4.32	4.27	100.629	3.34
			5	7	4	7	4	4		
	3.152	3.00	6.04	6.17	6.33	6.37	6.16	6.39	103.088	4.33
			2	8	9	5	9	3		
	6.00	6.00	8.97	9.04	9.05	9.14	9.17	9.03	97.446	2.46
			5	3	6	6	7	2		

This information has been added to line 231-234 and Supplementary Table 4 in the revised manuscript.

REVIEWERS' COMMENTS

Reviewer #1 (Remarks to the Author):

The author has responded appropriately to the reviewer's questions, either by appropriate experiments or by citing previous papers. Therefore, the reviewer considers that this study is a sufficient level for publication.

Reviewer #3 (Remarks to the Author):

The authors have followed up on my criticism. I have no further comments.

REVIEWER COMMENTS

Reviewer #1 (Remarks to the Author):

The author has responded appropriately to the reviewer's questions, either by appropriate experiments or by citing previous papers. Therefore, the reviewer considers that this study is a sufficient level for publication.

Response: Thank you very much for your recognition of our work! It is an honor for us to receive your appreciation.

Reviewer #3 (Remarks to the Author):

The authors have followed up on my criticism. I have no further comments.

Response: Thank you very much for your recognition of our work! It is an honor for us to receive your appreciation.